🔓 | **Open Peer Review** | Environmental Microbiology | Research Article

# Image-based physical characterization of magnetotactic bacteria from an environmental sample

Mara Smite,[1] Mihails Birjukovs,[1] Mila Sirinelli-Kojadinovich,[2] Sandrine Grosse,[2] Agnese Kokina,[3] Janis Liepins,[3] Dita Gudra,[4] Megija Lunge,[4] Davids Fridmanis,[4] Mihaly Posfai,[5] Andrejs Cebers,[1] Damien Faivre,[6] Guntars Kitenbergs[1]

**ABSTRACT** Magnetotactic bacteria (MTB) are a diverse group of microorganisms that are able to biomineralize magnetic nanoparticles. Most MTB remain uncultured, making population-level characterization of samples from natural environments difficult. The article reports on the discovery of a new and diverse MTB-rich site in the Ogre River, Latvia, and presents an integrated approach that combines 16S rRNA gene sequencing, transmission electron microscopy, and two novel open-source, automated, image-based physical methods to characterize bacteria populations within environmental samples. A pipeline for cell velocimetry is introduced, which uses a static magnetic field and a method to classify cell populations based on their magnetic moment using a modified U-turn method, where cell behavior is studied in an alternating magnetic field. This study demonstrates that image-based analysis methods provide a powerful, fast, and robust tool set for the analysis of MTB populations in complex environmental samples.

**IMPORTANCE** Magnetotactic bacteria are microorganisms that can be controlled using a magnetic field. They have potential applications in medicine, robotics, and environmental engineering, yet most species remain uncultured and poorly characterized. In this study, we report the discovery of a new MTB-rich site in the Ogre River, Latvia, and introduce open-source image-based methods that enable rapid and automated analysis of MTB directly from environmental samples. This accessible toolkit expands the methods to study MTB ecology and diversity, offering a robust framework for future ecological and applied microbiology research.

**KEYWORDS** U-turn, environmental biology, magnetotactic bacteria, magnetic moment

Magnetotactic bacteria (MTB) are a diverse group of microorganisms both phylogenetically and morphologically (1–4). They stand out from other prokaryotes due to their ability to biomineralize magnetic nanoparticles within their cells. In fact, these bacteria contain magnetosomes: single-domain magnetite and/or greigite crystals, surrounded by a protein-containing phospholipidic membrane (5, 6). The magnetosomes are typically ordered in chains along the easy axis of magnetization, which corresponds to the direction of magnetic moment (7–10). Magnetosomes are hypothesized to enable the passive alignment of bacteria with the Earth's magnetic field (MF) (11) and to help bacteria navigate complex environments (12). This characteristic can also be used to control the orientation and motion of cells in arbitrary directions using an external MF (13–17).

Due to the intrinsic magnetic properties, which allow external control of the cellular orientation; the flagellar apparatus, which drives the cellular motility; and the chemical toolbox, which allows cell functionalization with drugs, MTB are being studied for their applications in various fields. These include medicine (18–21), microrobotics (22–24), environmental engineering (25), and others (11, 26). All known MTB are Gram-negative

Address correspondence to Mara Smite, mara.smite@lu.lv.

The authors declare no conflict of interest.

(27), but can show considerable diversity in other aspects of their biology (28). MTB have been found in the *Alphaproteobacteria*, *Gammaproteobacteria*, *Etaproteobacteria*, and *Deltaproteobacteria* classes of the phylum *Proteobacteria*, in the phylum *Nitrospirae* (2, 8), and in the candidate phylum *Omnitrophica* (29). This leads to considerable variation in the properties of MTB that can be determined using microscopic methods: cell and magnetosome morphology and their magnetic response (28, 30). MTB of various shapes, including rod, spirillum, and cocci, have been imaged (30). Furthermore, iron-bearing magnetosome crystals can have various shapes and can be arranged in single or multiple chains, in clusters, or without order within the cell (28).

Despite their natural abundance, the vast majority of MTB remain uncultured under laboratory conditions (31, 32), and research on the diversity of magnetotactic microorganisms remains ongoing (33). The study of MTB from environmental samples provides information on their ecological roles, impact on biogeochemical cycles, diversity, and adaptations to specific environmental conditions, as well as insights into the discovery of new species with the potential to be cultivated in the laboratory. During the first expedition in Latvia, we explored three freshwater sites and found one location on the Ogre River where MTB were abundant. During our fieldwork in Latvia, we identified an MTB-rich freshwater location on the Ogre River. The collected samples contain an unknown mixture of MTB species and non-magnetic bacteria.

To study such complex environmental samples, we have developed a set of open-source image-based physical characterization tools that enable automated analysis of environmental MTB populations. These tools form the core methodology of this study and are applicable to both environmental and cultured samples. The central components of our approach include automated detection and trajectory reconstruction; an improved Bean model integrated with an automated U-turn analysis framework, capable of resolving multiple coexisting MTB populations based on magnetic behavior; and a novel velocimetry method requiring only a conventional optical microscope and a permanent magnet, making it highly accessible to laboratories without specialized equipment. We present a methodology that allows environmental samples containing a mixture of species based on the magnetic properties and velocity data of the cells to be analyzed, without the need of TEM or genetic analysis, thereby expanding the toolkit for magnetotactic microbiology and ecological biophysics. This is especially useful in field conditions or when a rapid characterization of a complex sample is necessary.

In this paper, we combine these automated physical methods with 16S rRNA metagenomic sequencing to explore taxonomic diversity and with transmission electron microscopy (TEM) to characterize the morphology of both cells and magnetosomes of a MTB-rich natural sample. Although one goal is to evaluate whether the results converge, the methods are complementary due to their different focus (genetic, morphological, and behavioral) and scales (population vs single-cell). This combined approach allows us to distinguish and analyze different MTB populations within a heterogeneous environmental sample based on the physical behavior of bacteria, the diversity of species, and their ecological organization.

## MATERIALS AND METHODS

### Sample collection

Environmental samples were collected during two expeditions: the first (Expedition A), conducted in August 2023, focused on sampling three freshwater sites to identify potential MTB habitats, while the second (Expedition B) (October 2023) aimed to resample the single location in the Ogre River identified as a source of various MTB. During Expedition A, 24 samples were collected in total, with at least seven from each freshwater body. Glass jars with metal lids were used as collection containers. The samples were obtained by manually filling the containers with a minimum of 1/4 of the sediment and the remaining volume with water. The sampling was carried out at depths up to 1 m. The collected samples were stored at an ambient temperature

of approximately +20°C. Expedition B resulted in five samples collected under similar conditions. The composition of the MTB population is likely strongly influenced by the sampling season, as water level and temperature changed significantly between the two expeditions.

## Bacterial extraction

The extraction of magnetotactic bacteria was based on the method described by Lefevre and Bazylinski (34). Before extraction, the contents of each container were thoroughly mixed by shaking them in a circular motion. Subsequently, the container was placed between two 1.2 T magnets with the north and south poles oriented in opposite directions. The magnets were placed a few centimeters above the sediment line and away from the wall of the jar. The sample was left undisturbed for 2 h, which did not yield results. Then, it was mixed again and left to settle for 12 h. If MTB are present in the sample, they tend to aggregate near magnets, forming a discernible pellet visible to the naked eye. The final step in the extraction process involved careful removal of the pellet from the inner side of the container with a pipette and its transfer to a 1.5 ml microtube for further analysis.

This technique yields approximately 30 µl of concentrated bacteria suspension. Because this is a natural sample, a portion of the bacteria within the sample may not exhibit magnetic properties. Following the initial extraction of the pellet, the process can be repeated until pellet formation ceases, indicating the death or exhaustion of magnetic organisms in the sample.

## Magnetic property confirmation

After extraction of the MTB, the magnetic properties of the sample were estimated using a slightly modified hanging drop method (28, 35, 36), where the cells are observed under a microscope in a thin droplet. A Leica light microscope with 40× magnification was used to observe the cells. The microscopy slides were prepared by the following steps: a 14 mm diameter hole was cut into a thick double adhesive tape, which was then placed on a standard microscopy slide, thus creating a well-like structure. Several microliters of the bacterial suspension were pipetted into the center of the well, ensuring that the edges of the droplet did not touch the tape. The sample was then sealed with a glass cover slip.

To find MTB, a 1.2 T permanent neodymium magnet was placed on the side of the microscope stage, with the south pole adjacent to the objective. The magnetic field diminishes with distance, and the field that interacts with cells is estimated to be around 10 mT, depending on the magnet placement. Most bacteria are expected to be north-seeking, as they come from a source located in the northern hemisphere; therefore, they would be attracted to the S of the magnet and concentrate at the edge of the droplet (Fig. 1b).

The magnet was then flipped, positioning the north pole toward the objective, causing MTB to follow the south pole and swim toward it. In contrast, if magnetic debris particles are present in the sample, they will rotate when the magnet is flipped but will not exhibit active swimming behavior, distinguishing them from motile microorganisms.

As a result of the first expedition, we found that the location of the Ogre River is an abundant natural source of MTB. No MTB was found in the other two locations.

## Optical microscopy

To analyze cell properties, a bacterial suspension was prepared using the previously described method. Measurements were performed with an optical microscope (Leica DMI3000B) equipped with a 40× magnification objective. Images were recorded using a Basler AC1920155UM camera, with a frame rate ranging from 20 to 100 frames per second.

A magnetic coil system was used to control the MF surrounding the sample. The MF was generated by three pairs of coils arranged around the sample stage in a

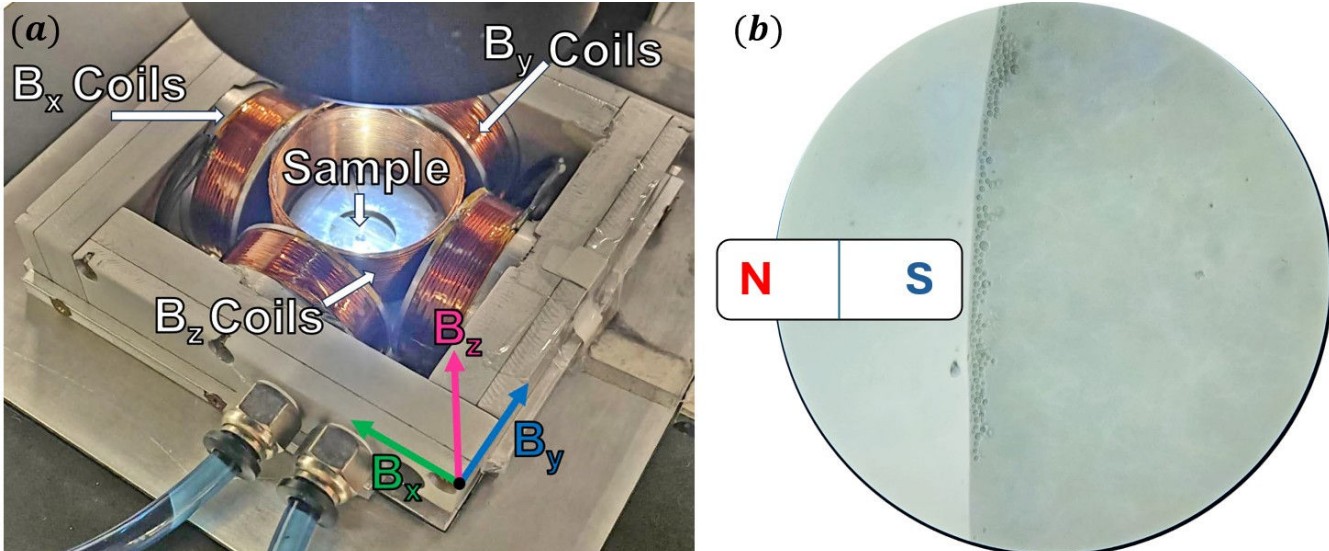

**FIG 1** Experimental setup: (a) the coil system used to generate magnetic field and (b) north-seeking bacteria concentrated at the edge of a hanging drop assay using a magnet.

configuration that allows control of a homogeneous field in three dimensions (Fig. 1a). The coil temperature was kept constant using a Huber Ministat 230 recirculation chiller.

Each pair of coils was controlled via an NI DAQ data acquisition card and LabVIEW software. The coils are powered by a bipolar power supply (Kepco BOP20-10M). The magnetic field strength is controlled by the current applied to the coils, where 1 A supplied to the coil results in a 1.7 mT strong magnetic field at the sample location (37, 38). Velocimetry experiments involved a constant 0.255 mT MF along the x-axis, and magnetic moment measurements used an alternating 0.255 mT square-wave MF.

## Image analysis

An algorithm was developed which accounts for static and dynamic image artifacts, stuck and motionless MTB, field-of-view translation and (re-)focusing, uneven illumination, image noise, and bacteria shape metrics. This is accomplished through a series of image histogram and background corrections, feature-preserving denoising, and non-linear contrast-boosting filters for MTB detection (39, 40). The MTB tracking was performed using the MHT-X code (40–42), an offline multiple-hypothesis tracking implementation with a graph-based framework based on Algorithm X (43), incorporating physics-based motion prediction models and support for system boundary conditions, flow field data, and more. The underlying methodology is described in detail in reference 44, and the code is open-source (see "Data availability").

## Velocimetry

To assess the velocity statistics for the sample, we applied a constant MF to the sample, and the motion of the bacteria was imaged using the light microscope. Using MF, cells were concentrated next to a capillary wall for several minutes; then, MF was reversed, resulting in cell movement away from the wall. The recording continued until most of the cells had swum out of the field of vision. For both magnetic moment measurements and velocimetry, MTB trajectories were reconstructed using the MHT-X tracking algorithm (40–42), which is open-source (see "Data availability"). Velocity and other track properties are readily derived from reconstructed tracks.

## Magnetic moment calculation

Cell magnetic moment measurement is performed using the standard U-turn method—a pulsed MF induces alternating U-turn-like motion of the bacteria (45). The U-turn width encodes the information about the MTB magnetic moment. To derive magnetic moment statistics and scaling with MTB size from the acquired trajectories, we use our explicit, fully automatic method developed and previously used for MSR-1 MTB (44). Trajectories are first decomposed into U-turns, which are then fitted to a theoretical U-turn shape function derived in reference 44:

$$y = -\frac{L}{\pi}\ln\left(\sec\left(\frac{\pi x}{L}\right)\right) \qquad (1)$$

where $L$ is the asymptotic U-turn width and $x$ and $y$ are the coordinates. When $L$ is determined, the cell magnetic moment $m$ can be determined as follows (44):

$$m = \frac{\pi \alpha \upsilon}{HL} \qquad (2)$$

where α is the MTB rotational drag coefficient with $\alpha = 8\pi^2 \eta \cdot R^3$, η is the viscosity of the medium, $R$ is the effective radius of the microorganism, and $H$ is the strength of the applied pulsed MF. Here, $\eta = 0.90 \cdot 10^{-3}$ Pa $\cdot$ s, the pulsed MF is set to $B = 2.55 \cdot 10^{-4}$ $T$, and $R$ is determined for each bacterium as a trajectory average. Since MTB in the sample have various shapes, effective radius is used to simplify the drag coefficient calculation and compare MTB on a more equal basis. $R$ is calculated by first converting the detected cell area to a circle of equal area, and then the effective radius is found.

The code for determining $m$ from MTB trajectories is open-source (please see "Data availability").

## MTB population identification

Population distributions are considered symmetric Gaussians $G(\mu_1, \mu_2, \sigma_1, \sigma_2, \rho, a)$ with means $\mu_{1,2}$, variances $\sigma_{1,2}$, covariance ρ, and scales $a$. An optimization sweep is performed, where a different number of Gaussians $n \in N$ is used to fit the data by optimizing the parameters for all $G(\mu_1, \mu_2, \sigma_1, \sigma_2, \rho, a)$ using differential evolution. The optimal n with the best fit is then given by the minimum Akaike information criterion (AIC) (46). The significant populations are then selected based on their contributions to the PDF integral volume reconstruction.

## Transmission electron microscopy (TEM)

To characterize both morphology and magnetosomes, transmission electron microscopy (TEM) was used. Samples were prepared by depositing bacterial suspensions, prepared using the method described in "Bacterial extraction," onto carbon-coated copper TEM grids. The suspensions were then air-dried. The images were collected using a Tecnai G2 BioTWIN (FEI Company) electron microscope equipped with a charged-coupled device (CCD) camera (Megaview III, Olympus Soft Imaging Solutions GmbH) at an accelerating voltage of 100 kV and with a Talos F200X (Thermo Fisher) TEM operated at 200 kV.

## DNA extraction and sequencing of the 16S rRNA gene V3-V4 region

The DNA of the samples was isolated using FastDNA SPIN Kit for Soil (MP Biomedicals, USA) according to the manufacturer's guidelines. The quantity of the extracted DNA was assessed using the Qubit dsDNA HS Assay Kit on a Qubit 2.0 Fluorometer (Thermo Fisher Scientific, USA). The two-stage PCR protocol was applied for MiSeq library preparation. 341F and 805R primers were designed for the PCR amplification of the 16S rRNA gene V3–V4 region, which is specific to the domain bacteria, and contained Illumina overhang adapters (47). Microbial DNA (4 ng) was amplified separately using V3 and V4 primers

with Phusion U Multiplex PCR Master Mix (Thermo Fisher Scientific) under the following reaction conditions: denaturation at 98°C for 30 s, 35 cycles of 98°C for 10 s, 67°C for 15 s, 72°C for 15 s, and fragment elongation at 72°C for 7 min. The yield of the acquired PCR products was assessed by 1.2% agarose gel electrophoresis and purified using a NucleoMag NGS Clean-Up and Size Select Kit (Macherey-Nagel, Germany). The concentration of PCR products was measured using a Qubit dsDNA HS Assay Kit and a Qubit 2.0 Fluorometer, and the samples were normalized to 4 ng/µl. During the second PCR stage, Illumina MiSeq i7 and i5 indexes were added to the 4 ng of the V3 and V4 PCR product using custom-ordered Nextera XT Index Kit (Illumina Inc., USA) primers (Metabion International AG, Germany). For this reaction, Phusion U Multiplex PCR Master Mix was used under the same thermal cycler reaction conditions as specified for the first PCR stage. The 16S rRNA PCR products were then pooled and purified for the sequencing reaction using NucleoMag magnetic beads. The quality and acquired amount of the 16S rRNA V3–V4 amplicons were assessed using an Agilent High Sensitivity DNA Chip kit and Agilent 2100 BioAnalyzer (Agilent Technologies, USA) and a Qubit dsDNA HS Assay Kit and Qubit 2.0 Fluorometer, respectively. Before sequencing, all samples were pooled at equal molarities and diluted to 6 pM. The samples were paired-end sequenced using the 500-cycle MiSeq Reagent Kit v2 on an Illumina MiSeq (Illumina Inc., USA). After the sequencing run was completed, the individual sequence reads were filtered by MiSeq software to remove low-quality sequences. All MiSeq quality-approved, trimmed, and filtered data were exported as fastq files.

## 16S rRNA gene sequencing data analysis

Sequence reads were quality filtered using Trimmomatic v.0.39 (48) with a leading quality of Q20, a trailing quality of Q20, and sequences shorter than 36 nucleotides were discarded. All quality-approved sequences were imported into the QIIME 2 v.2021.11 (49) environment for further analysis. The DADA2 (50) plug-in was used to pair forward and reverse reads, as well as for additional sequence quality control and chimeric sequence removal using a pooled consensus method. The resulting feature table and sequences were used for *de novo* clustering employing the VSEARCH plug-in with a 97% identity threshold (51). Thereafter, *de novo* multiple sequence alignment was performed using the MAFFT method (52), while phylogenetic trees were constructed using FastTree 2 (53). *De novo* clustered sequences were used for taxonomic assignment with the pre-fitted sklearn-based (54) taxonomy classifier, based on the SILVA v.132 97% identity reference database that was trained with the naïve Bayes classifier.

## RESULTS AND DISCUSSION

Two field expeditions were conducted in Latvia, Madona municipality (Fig. 2) with the aim of identifying and characterizing MTB. Expedition A was conducted in August 2023 with the objective of finding a source of MTB and characterizing their morphology using TEM. Three locations were sampled: Lake Mezezers (coordinates [56.662073, 25.718520]), Lake Kala (coordinates [56.863128, 25.810166]), and Ogre River (coordinates [56.895449, 25.641971]), with samples from the Ogre River containing abundant and morphologically different MTB. The second expedition, Expedition B, was conducted in October 2023, revisiting the Ogre River site with the aim of obtaining more biomass to further characterize the diversity of MTB found there.

In Expedition A, a total of 24 samples were collected from three different freshwater environments. After MTB extraction, two of the containers that originated from Ogre River revealed the presence of magnetic cells, as confirmed by optical microscopy. TEM images were later produced from the samples acquired during this expedition. Without enrichment, the lifespan of MTB in the sample containers did not exceed 2 days, which means that no MTB sample could be extracted from the jar after a 48-hour period. Expedition B further explored the diversity of MTB encountered in the Ogre River, yielding an MTB sample volume sufficient to perform a 16S rRNA analysis, as well as to

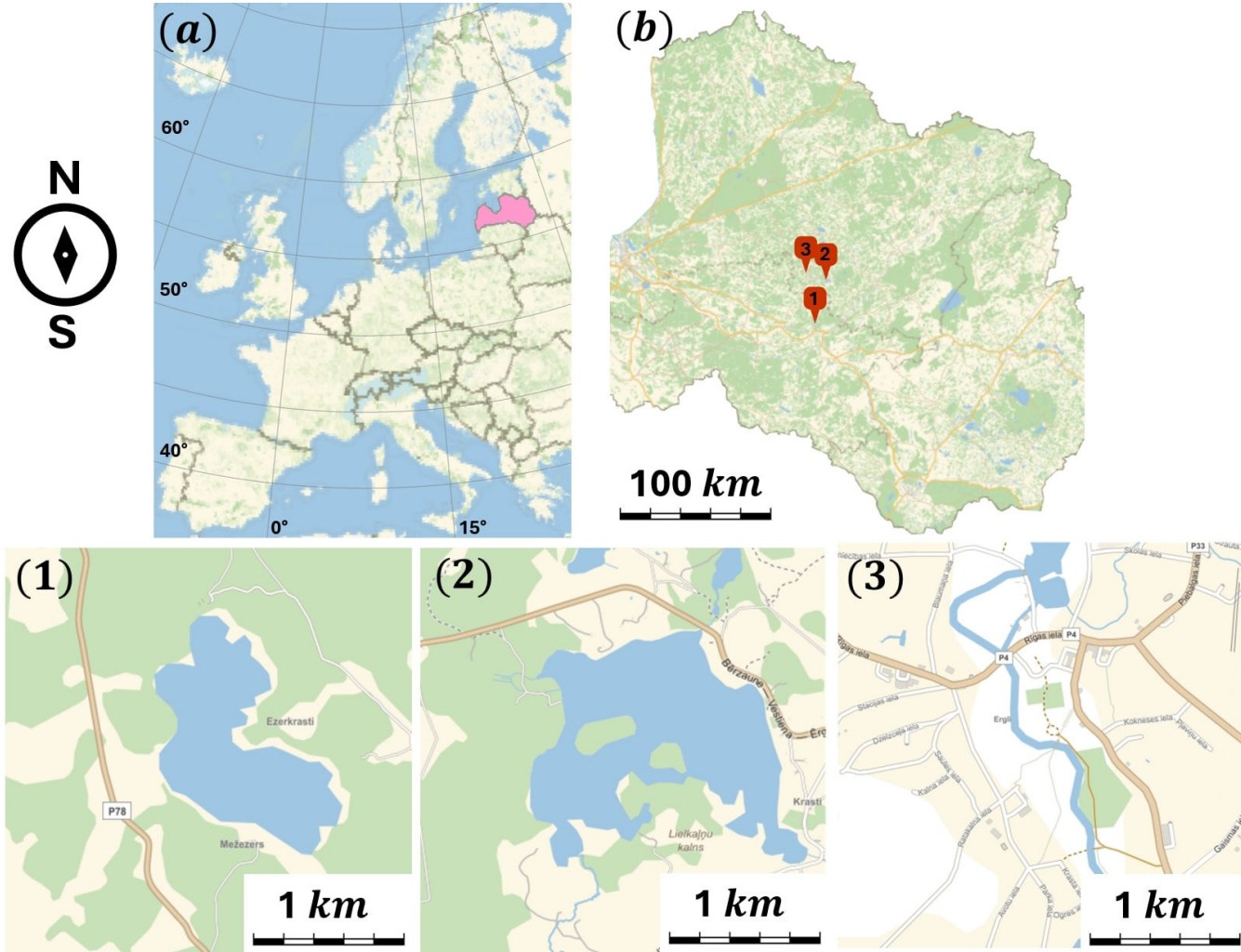

**FIG 2** MTB sampling locations: (a and b) map of Latvia. Expedition A sampling locations 1 (Lake Mezezers), 2 (Lake Kala), and 3 (Ogre River) in search of MTB. Expedition B collected MTB from location 3 for further analysis.

characterize the population using our proposed automated methods of physical cell characterization. An overview of the data collected is found in Table 1.

## Morphological and genetic diversity

The MTB found in the Ogre River show considerable diversity in cell morphology, as well as in magnetosome crystal shape and in the cellular localization and organization of crystals (Fig. 3). Different cocci and rod-shaped bacteria were observed in TEM, as well as cell aggregates. Some cells had small attachments with magnetosomes, as seen in Fig. 3a. It might be a separate, smaller MTB attached to the cells, or minicells formed as a

**TABLE 1** An overview of data sets collected during the search for MTB in Latvia

| Expedition | Results |
| --- | --- |
| A | TEM images of cells (see "Morphological and genetic diversity") |
| B | 16S rRNA analysis (see "Morphological and genetic diversity") |
| B | Velocimetry data set taken on day 1 after sampling (see "Velocimetry-based characterization") |
| B | Velocimetry data set taken on day 2 after sampling (see "Velocimetry-based characterization") |
| B | Magnetic moment data set (see "Magnetic moment-based characterization") |

result of abnormal cell division (55). The cell aggregates typically consist of two to three larger bacteria and several smaller ($d < 1$ μm) bacteria attached to the main cluster (Fig. 3c). TEM images show that some cocci aggregates possess numerous flagella, although the cell clustering could be formed as an artifact of the TEM procedure.

Rod- and spiral-shaped bacteria of various sizes were found, and TEM images show that, in some cases, they possess a single flagellum. For some cocci, flagellar tufts could be seen in TEM, suggesting a bilophotrichous structure, which was also confirmed by the zigzag movement pattern characteristic of the helical motion (34) observed in some cells observed by optical microscopy (Fig. 8).

The crystal shapes of magnetosomes in samples from the Ogre River also show considerable diversity: cubooctahedral magnetosomes (Fig. 3a), bullet-shaped (Fig. 3b),

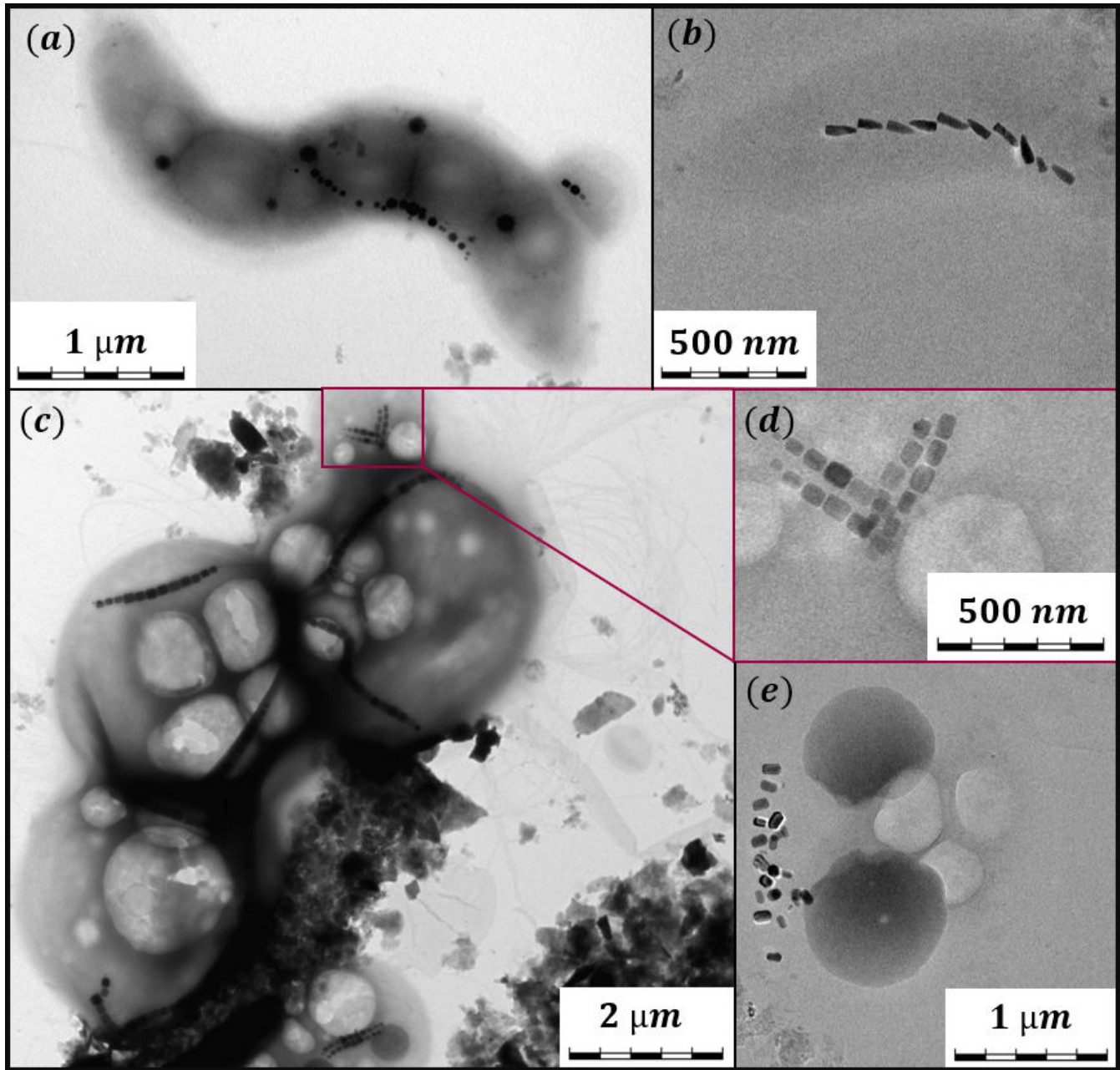

**FIG 3** TEM images of diverse MTB found in the Ogre River: (a) spirillum; (b) MTB with bullet-shaped magnetosomes; (c) a cell aggregate of cocci with 2 magnetosome chains in a single cell; (d) a double chain of prismatic magnetosomes; and (e) a partly disordered magnetosome configuration.

elongated prismatic (Fig. 3c and d), and "ladyfinger"-shaped (56) (Fig. 3e). Magneto-somes can be configured in a single chain per cell (Fig. 3a and b), two opposite chains in a cell (Fig. 3c), two double chains per cell (Fig. 3c and d), and a partly disordered cluster (Fig. 3e). Although some MTB sources can contain a single magnetic population, it is common to find mixed MTB species in an environmental sample (33, 35, 57).

Metagenomic diversity analysis of the Ogre River sample was performed using the Illumina MiSeq sequencing platform for 16S rRNA V3–V4 gene sequencing. The acquired data revealed a great diversity of microbiological organisms from eight families (Fig. 4). The proteobacteria—alpha, beta, and delta—were present in the sample, but the dominant genus was *Sphingomonas*, which amounted to 94.9% of all sequenced bacteria. These were followed by 4.7% of the sequencing reads coming from five different genera of the *Burkholderiaceae* family, while 0.1% of the read sequences were characteristic of the genera *Magnetococcus* and *Geobacter*. The genera *Sphingomonas* and *Magnetococcaceae*, as well as *Burkholderiaceae* family, are known to include MTB (58).

The genus *Sphingomonas* consists of species highly abundant in the soil and water and has been found in rivers before (59). *Sphingomonas* spp. are typically associated with nutrient- and oxygen-rich media and have a tendency to form biofilms together with other microorganisms (60). Many species are associated with plant roots and potentially also with soil effluxes to the river. However, there are results of the rich occurrence of *Sphingomonas* from deep sediments beneath Earth's surface in an oxygen-poor environment. Potentially, these anoxic strains convert ligninous substrates (61).

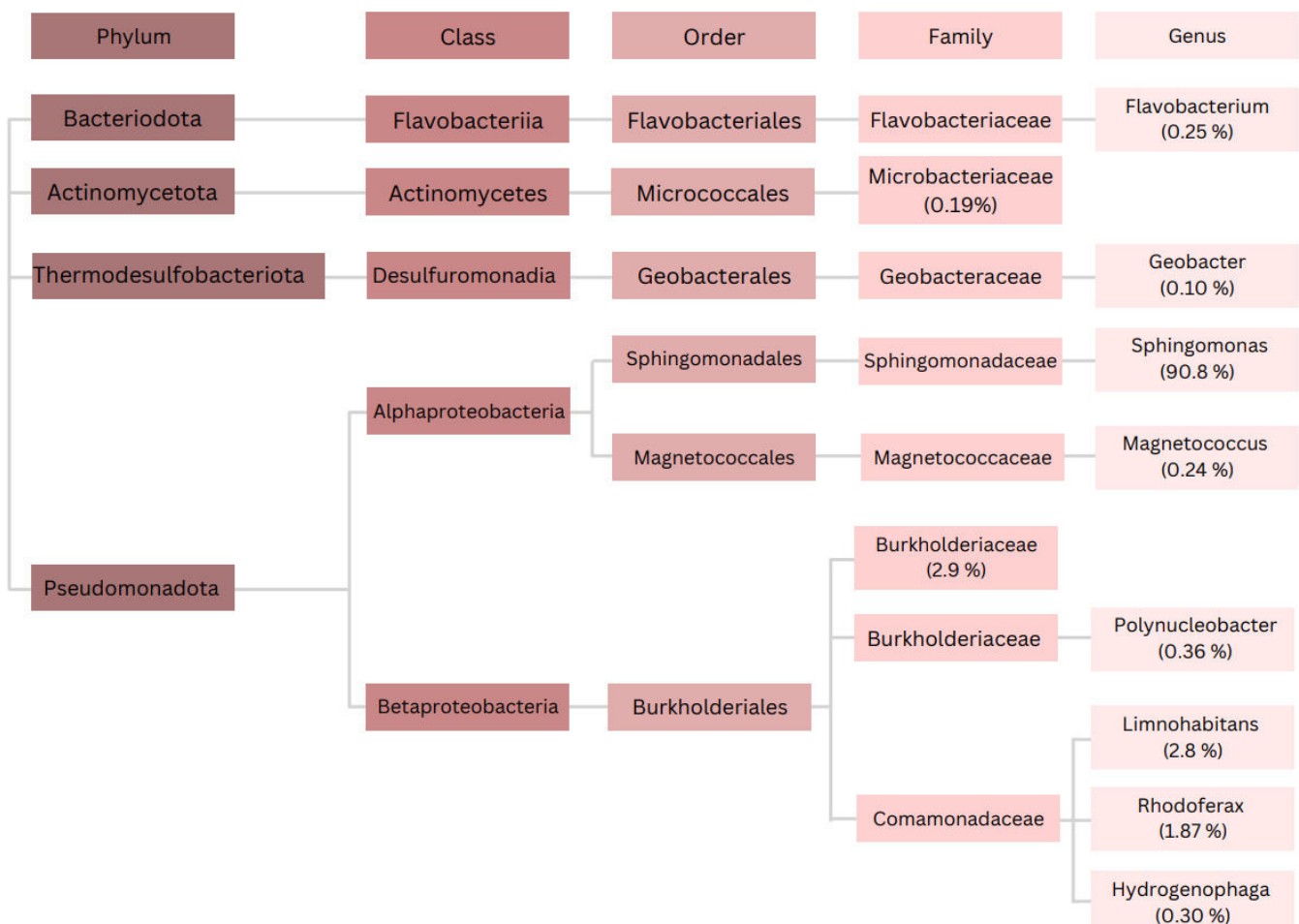

**FIG 4**  The relative abundance of bacteria found in the Ogre River sample. MTB have been previously found in *Sphingomona* and *Magnetococcus* genera, as well as *Burkholderiaceae* family.

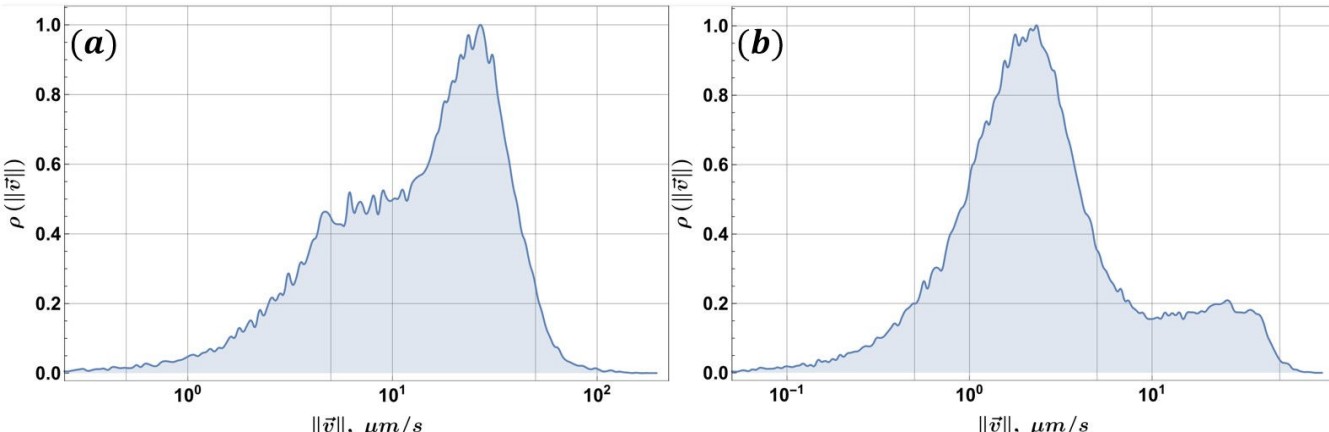

**FIG 5** Relative frequency ρ histogram for MTB velocity magnitude $\|\vec{v}\|$ in the $\log_{10}$ scale: samples from Expedition B: (a) day 1 and (b) day 2. Freedman–Diaconis binning is used in both cases.

The appearance of *Sphingomonas* spp. in the magnetotactic enriched microbial community of the Ogre River may be related to the "contamination" of the river from rich soils of surrounding plants. In addition, it is possible that *Sphingomonads* exist permanently in the Ogre River, regardless of the soil (62). Interestingly, a similar pattern has been reported in the floodplain of the Araguaia River, Brazil, where most of the discovered MTB also belonged to the *Pseudomonadota* phylum (33).

Although we did not characterize the species abundance of non-enriched river bed samples, we presume that the initial sampling could have disrupted a microbial mat, and over time (48 h), motile bacteria could migrate out from it and evenly diffuse throughout the sample. When quantifying the abundance of sample species, we relied on the molecular methods (DNA extraction, PCR, and Illumina sequencing), which are not biased toward species richness. From the point of view of magnetic response, most probably *Sphingomonas* spp. form the population with a velocity of 0–10 µm and a radius of approximately 0.5 (see Fig. 9) (63).

*Geobacter* bacteria can reduce Fe (III) in an anaerobic sedimentary environment, but are not known to form magnetosomes (64). The metagenomic diversity confirms that the Ogre River is an environment where MTB can thrive, with an unknown number of species co-existing at the sampling site but belonging to at least three different families. As is typical in environmental samples, non-magnetic bacteria were also detected. The reads from the *Flavobacteriaceae* and *Microbacteriaceae* families amounted to 0.1% of the entire repertoire each, but MTB have not yet been found in these genera, especially *Microbacteriaceae*, as they are Gram-positive and all MTB identified so far are Gram-negative (27).

## Velocimetry-based characterization

In addition to standard TEM and gene sequencing methods, we developed an open-source velocimetry data processing method (see "Data availability"), based on conventional optical microscopy and a static magnetic field. In general, for data acquisition, this method only requires a microscope equipped with a camera and a small neodymium magnet as the MF source. The static magnetic field ensures that MTB are concentrated on the capillary boundary, and when it is reversed, MTB will follow the new MF direction. Long trajectories can then be recorded that span the whole field of view. The MTB was tracked, and their velocities (Fig. 5) and effective radii were extracted using automated tools. The sample size for each of the subfigures in Fig. 6 is ≈ 75K instances of radius–velocity pairs. Multi-Gaussian fits, constrained by the Akaike information criterion (AIC) (46), were used to determine how many cell populations were present in the natural sample (Fig. 6; the population parameters are summarized in Table 2; all populations are

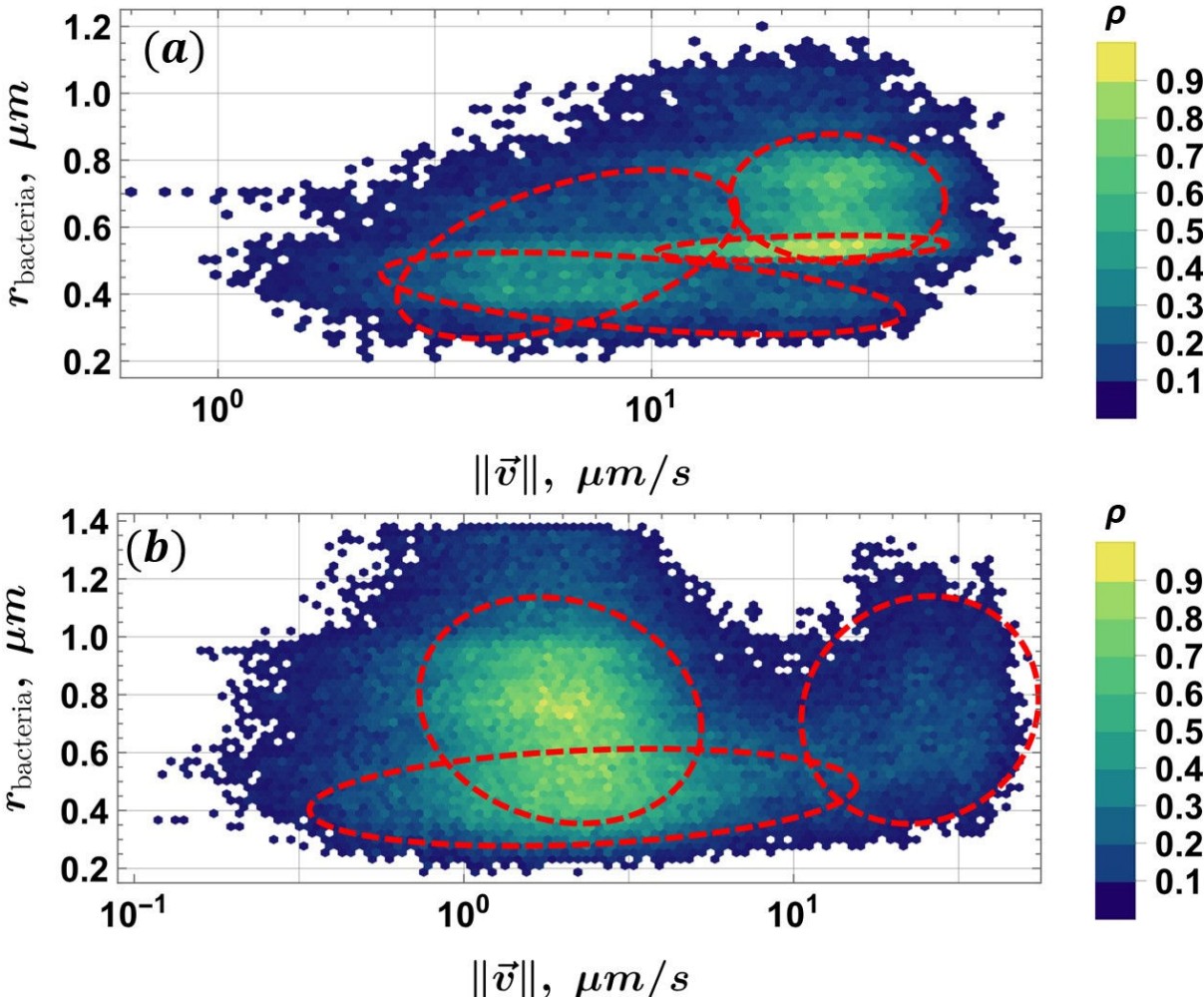

**FIG 6** Relative frequency histograms for MTB effective radii $r_{bacteria}$ and velocity magnitude $\|\vec{v}\|$ in the $\log_{10}$ scale, samples from Expedition B: (a) day 1 and (b) day 2. Freedman–Diaconis binning was used in both cases. Dashed ellipses denote populations determined using the AIC-constrained multi-Gaussian fit, and the color coding indicates the relative frequency. $N \approx 75,000$ for each data set.

shown together in Fig. 7). Note that these velocimetry results are not filtered for MTB compliance with the applied MF direction, ensuring inclusion of all detected motion; however, our code provides the option to filter out statistics for MTB not following the MF direction.

The relative frequency velocity histogram from Expedition B (Fig. 5a) suggests that several populations may be present in the sample, with the most probable velocity ~27 µm/s. Cell velocities up to 100 µm/s are observed. The same velocimetry experiment using a fresh sample from the container was performed the next day (Fig. 5b)—notice that the velocity distribution is now roughly bimodal, with the most probable velocity ~2 µm/s and another notable local maximum at ~26 µm/s.

Cell velocities under ~7 µm/s can be considered passive movement of MTB (65). The cell velocity range extends up to ~50 µm/s, which means that faster cells were not present in the sample during the measurement or had perished in the 24-hour period since the previous experiment. Note also that the most probable velocity in Fig. 5a and the secondary maximum in Fig. 5b have consistent values, possibly corresponding to the same population that has diminished over time. Velocity distributions are a tool that is better suited for monoculture samples, as they can be difficult to interpret when multiple unknown distributions are present in the sample. For cases with an unknown number of

**TABLE 2** Population parameters for velocimetry, days 1 and 2 (Fig. 6), with uncertainties[a]

| $\mu_1$, $\log_{10}\|\vec{v}\|$ ($\mu$m/s) | $\mu_2$, $r_{bacteria}$ ($\mu$m) | $\sigma_1$ | $\sigma_2$ | $\rho$ | $\|\vec{v}\|$ ($\mu$m) |
|---|---|---|---|---|---|
| Day 1 | | | | | |
| $0.98 \pm 0.01$ | $0.402 \pm 0.002$ | $0.40 \pm 0.01$ | $0.081 \pm 0.002$ | $-0.49 \pm 0.02$ | [3.8; 24] |
| $1.348 \pm 0.004$ | $0.5379 \pm 0.0003$ | $0.223 \pm 0.003$ | $0.0247 \pm 0.0004$ | $0.30 \pm 0.02$ | [13; 37] |
| $1.429 \pm 0.002$ | $0.686 \pm 0.001$ | $0.165 \pm 0.001$ | $0.127 \pm 0.001$ | $-0.046 \pm 0.012$ | [18; 39] |
| $0.806 \pm 0.006$ | $0.519 \pm 0.005$ | $0.26 \pm 0.01$ | $0.167 \pm 0.005$ | $0.51 \pm 0.03$ | [3.5; 12] |
| Day 2 | | | | | |
| $0.38 \pm 0.01$ | $0.439 \pm 0.001$ | $0.49 \pm 0.01$ | $0.103 \pm 0.002$ | $0.28 \pm 0.02$ | [0.78; 7.4] |
| $0.288 \pm 0.001$ | $0.751 \pm 0.003$ | $0.239 \pm 0.001$ | $0.265 \pm 0.02$ | $-0.16 \pm 0.01$ | [1.1; 3.4] |

[a]$\mu_1$, mean velocity magnitude $\|\vec{v}\|$ in $\log_{10}$ scale; $\mu_2$, mean bacterial radius; $\sigma_{1,2}$, standard deviations for $\mu_{1,2}$; $\rho$, correlation coefficient; $N \approx 75,000$ for each data set.

populations, as in an environmental sample, evaluating a relative frequency histogram that relates cell velocity and radii (Fig. 6) could be more appropriate.

Figure 6 shows the distribution of cells according to their effective radii $r$ and the corresponding velocity $v = \|\vec{v}\|$. For multi-Gaussian fits, AIC was used to penalize the parameter count of the Gaussian mixture model, and then, only the populations that contribute significantly to the reconstruction of the integral volume of the probability density function (PDF) are selected as relevant. In Fig. 6a, four significant cell populations are found in the sample: two with a positive correlation $r(v)$ and two with a negative

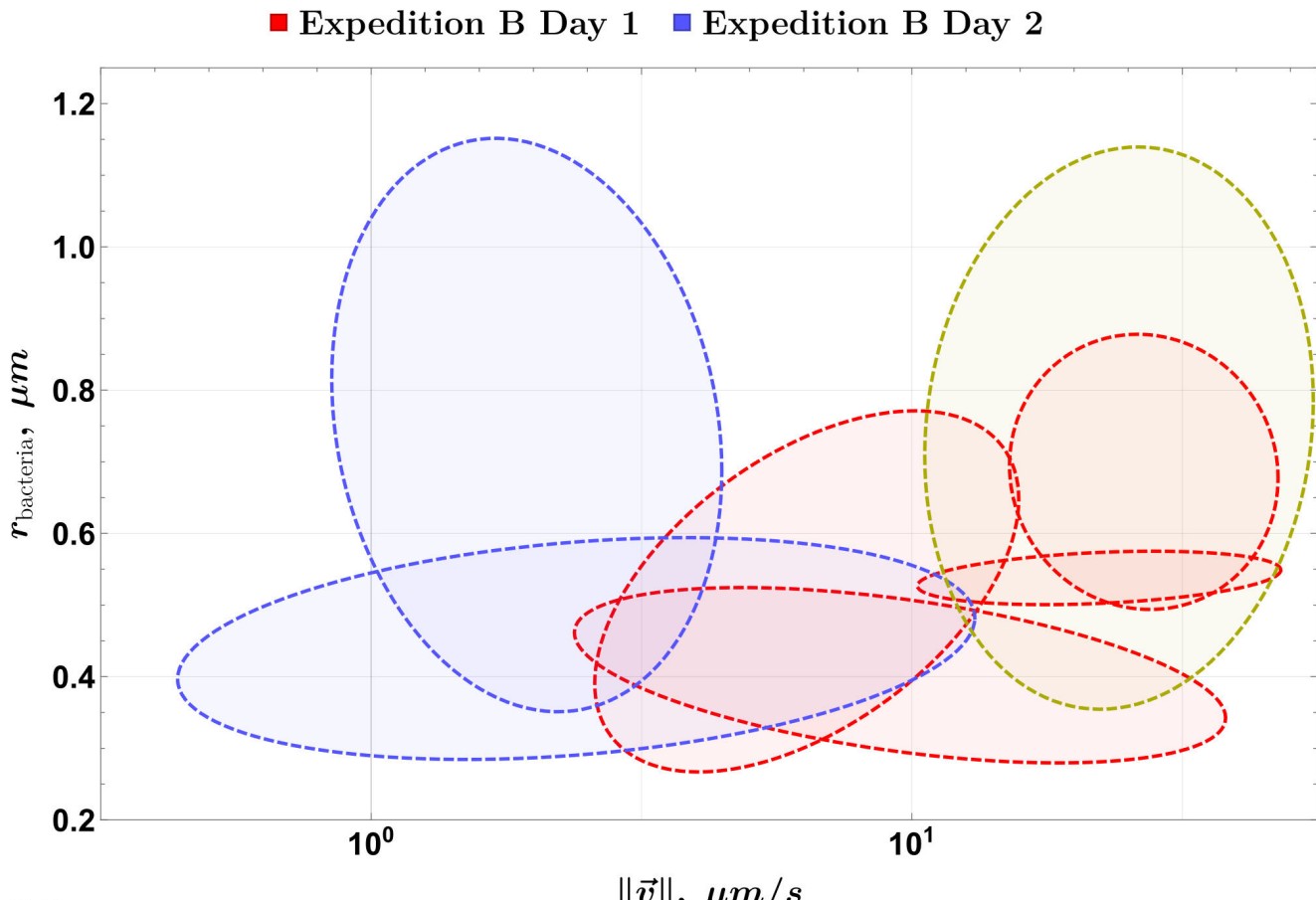

**FIG 7** Population outlines for both measurement days (legend at the top). The dashed green line denotes one of the detected populations from day 2 that is not significant.

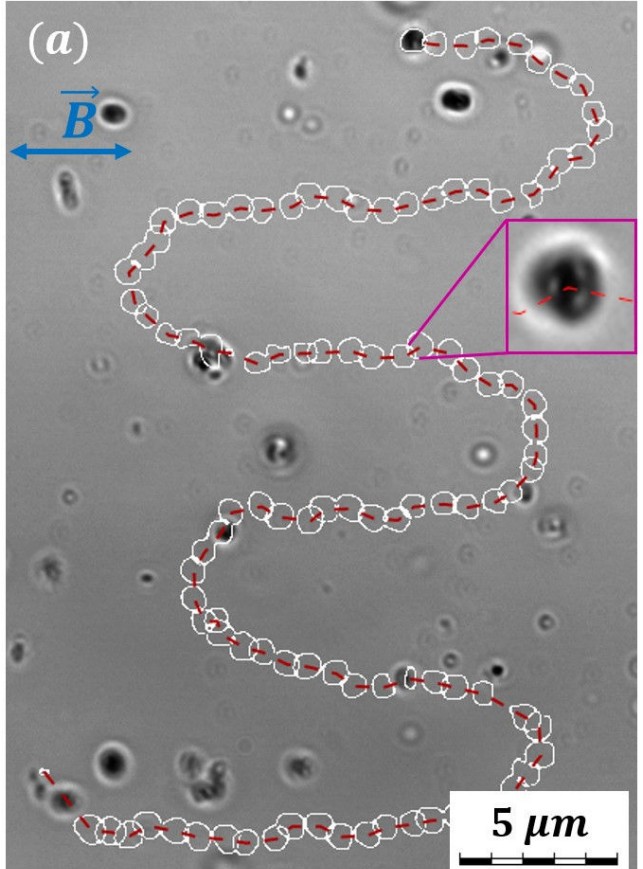 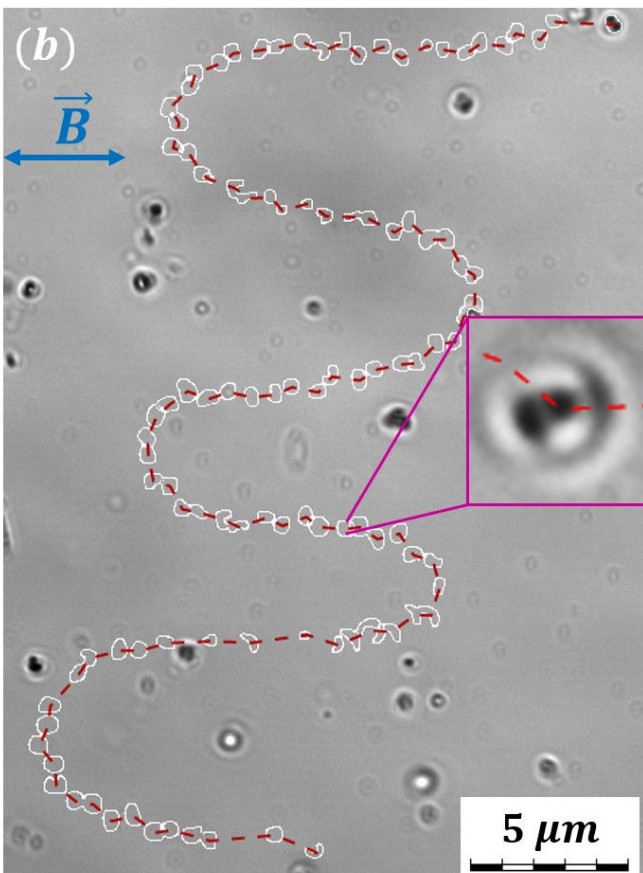

**FIG 8** Examples of cell trajectories (red dashed lines) in an alternating MF: (a) a coccus and (b) a diplococcus. White contours outline the cell shapes.

correlation. In contrast, in Fig. 6b, only two populations remain: one with a positive correlation $r(v)$ and the other with a slightly negative correlation. The cell population with radii in the range between r $\in$ ~ [0.2; 0.6] µm is present in the samples of both days.

The populations shown in Fig. 6 are combined in a single plot for easier comparison in Fig. 7. Two populations are steadily present on both days, but a third one, not statistically significant and denoted by a dashed green ellipse, appears on day 2. Its area overlaps with the two populations that are present in the day 1 sample, but not in day 2. This could mean that it is a remnant of the two populations, where most cells have perished, but enough remain alive to be detected.

## Magnetic moment-based characterization

Automatization of cell magnetic moment measurements is another powerful tool that allows us to categorize cells based on their magnetic properties and gives a deeper insight into cell behavior when used together with velocimetry. We used our automated U-turn method (44) to estimate the magnetic moment for 846 eligible individual tracks— example trajectories are shown in Fig. 8. Cells from Expedition B Day 1 were studied.

The previously described velocimetry method was applied to the data set, relating the MTB velocity and effective radius. In the distribution shown in Fig. 9, two significant velocity/radius populations are found in the measured sample, and the magnetic moment $m$ distribution is shown in Fig. 10b. The latter indicates that the most probable magnetic moment is $m = 1.5 \cdot 10^{-15} A \cdot m^2$, the mean magnetic moment is $\langle m \rangle = 2.06 \cdot 10^{-15} A \cdot m^2$, and its standard deviation is $\sigma(m) = 1.0 \cdot 10^{-15} A \cdot m^2$. For comparison, the MSR-1 MTB is $m$ ~$10^{-16} A \cdot m^2$ (44, 66–68), and for uncultured magneto-

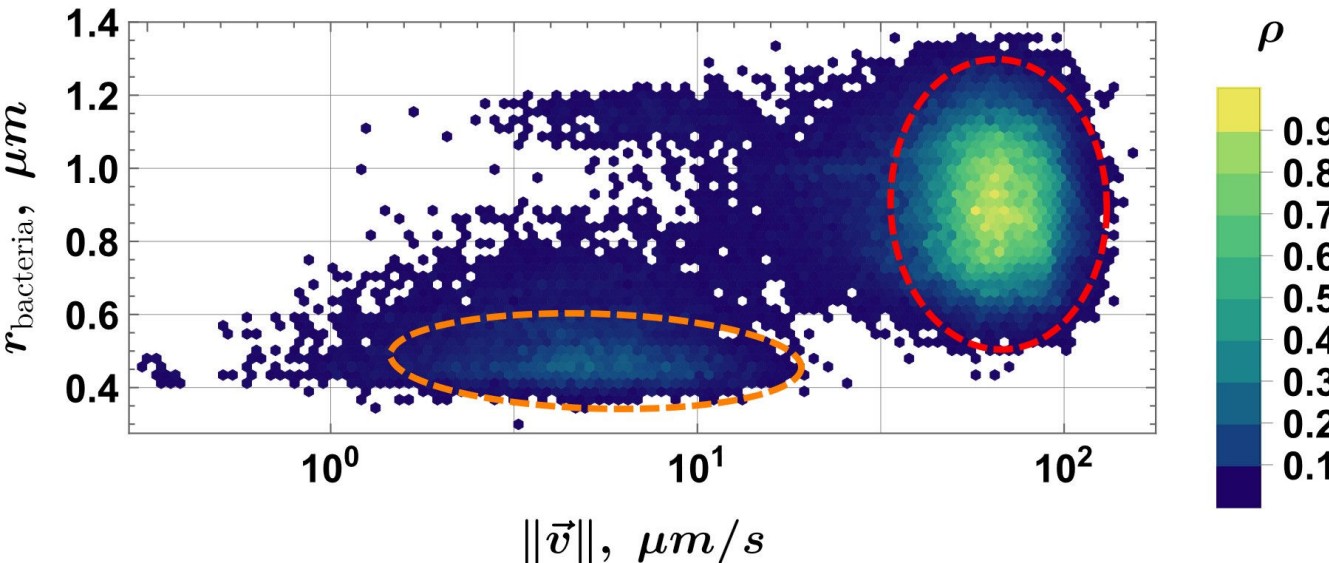

**FIG 9** Data obtained from magnetic moment *m* measurements via the modified U-turn method: velocitmetry: *v* and effective radius *r* distribution (Freedman–Diaconis binning), with significant populations marked with dashed ellipses—orange color for the non-magnetic population and red for the magnetic population.

tactic cocci, the magnetic moment obtained by the U-turn method was reported as $m = 8.2 \cdot 10^{-15}\,A \cdot m^2$ (69). $m \sim 10^{-13}\,A \cdot m^2$ has been reported for magnetotactic holobionts (70).

In Fig. 10a, an overall positive correlation can be seen between the effective radius of the cells and their magnetic moment. A critical radius threshold is identified at $r_c = 0.57$ µm using an error-weighted linear model (44), below which the cell size is insufficient to host magnetosomes that produce measurable magnetic moment *m*. Similar positive correlations between bacteria size and magnetic moment have been reported in MSR-1 (44) and in uncultured magnetic cocci (69).

A comparison between the velocimetry data (Fig. 6a) and the smooth density histogram of magnetic moments (Fig. 11) reveals that the population of smaller bacteria detected in velocimetry is absent from the magnetic moment measurements. This discrepancy likely arises from the presence of non-magnetic bacteria in the sample, which can be tracked during motion analysis but do not exhibit U-turn behavior in response to alternating MF. A more in-depth analysis outlined in the Appendix indicates that the population with $\|\vec{v}\| \in [18; 39]\ \mu\mathrm{m/s}$ and $r \in [0.56; 0.81]\ \mu\mathrm{m}$ in Fig. 6a (Table 2) is composed of smaller clusters (Fig. 14), where the global number density maximum corresponds to the maximum in Fig. 6a, despite the bias introduced by the filtering criteria applied in the moment calculation algorithm.

Data from the magnetic moment measurements can be further used to analyze the population composition of the sample. The analysis pipeline (see "Data availability") includes a built-in option to perform automatic multi-Gaussian fitting of the magnetic moment versus cell effective radius smooth density histogram. As with velocimetry, the number of populations is restricted by AIC, and only statistically significant populations are presented in the resulting figure.

As shown in Fig. 11, the optimal fit according to the AIC corresponds to three populations, differentiated by their magnetic properties. Notably, one of the populations in this model exhibits a negative correlation between the cell radius and the magnetic moment, a trend that, to the best of our knowledge, has not been reported previously. It is unlikely that this correlation arises from cellular aggregates (e.g., doublets or triplets), as the corresponding effective radii (0.86–0.93 µm) are too small to accommodate multiple cells. Thus, the presence of a distinct cluster cell population, as suggested in Fig. 3c, is not supported by the physical dimensions observed.

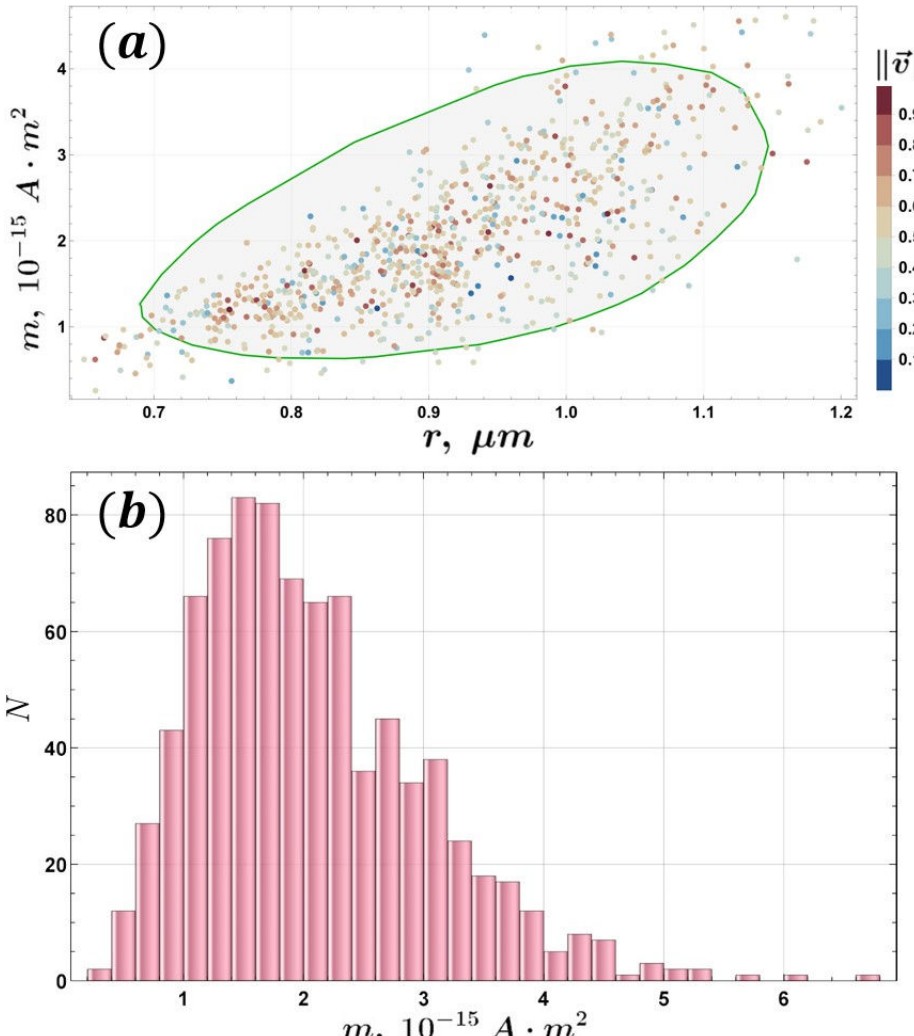

**FIG 10**  (a) Magnetic moment *m* versus MTB effective radius. The measured *m* values depending on the cell effective radius are represented by gray dots, with the *q* = 0.95 quantile uncertainty region indicated as the light gray area with a green boundary. The uncertainty region is derived by computing 250 directional quantiles (71, 72), connecting the quantiles with a smooth curve, and applying curve evolution polyline simplification (73) to the resulting quantile envelope. The data points are color-coded by their normalized relative velocity. (b) Magnetic moment histogram (count *N* versus *m*, Freedman–Diaconis binning, *N* = 846).

Furthermore, while holobionts have been reported to exhibit magnetic moments (70) on the order of $1.8 \pm 0.8 \cdot 10^{-13}\ A \cdot m^2$, such values are typically associated with protists that carry magnetotactic symbionts and have overall dimensions in the range of 10–20 µm. These dimensions are incompatible with the cell sizes observed in our study. Consequently, we interpret the three populations identified to represent three separate species, each represented at the single-cell level in the experimental analysis. It is important to note that a negative correlation is not set as a constraint in the Gaussian fitting process because it has not been strictly ruled out as impossible. The population parameters are shown in Table 3.

## Conclusions and outlook

We identified a previously undocumented source of MTB in the Ogre River, Latvia. Using a combination of TEM and 16S rRNA sequencing, and newly developed automated image-based physical methods, we characterized the diversity and magnetic behavior

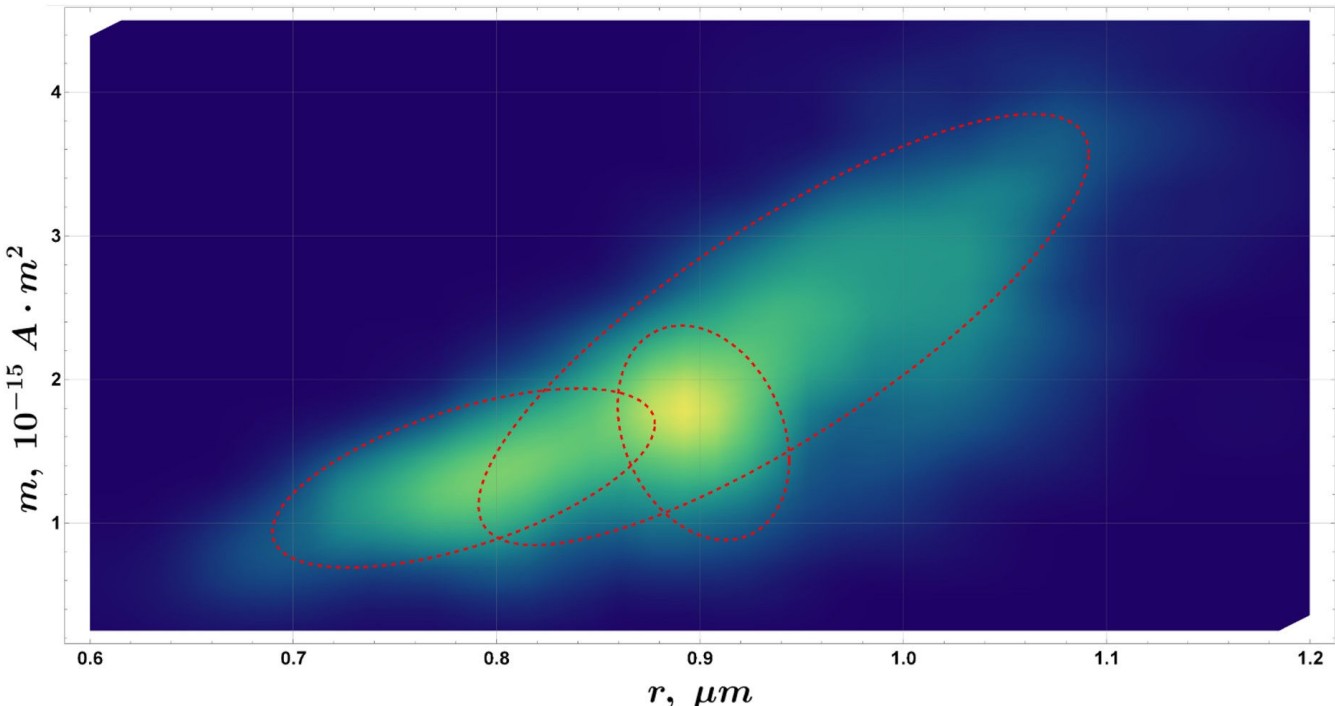

**FIG 11** Magnetic moment *m* versus MTB effective radius *r*: ansmooth density histogram (Sheather-Jones bandwidth estimator, Epanechnikov kernel). Dashed ellipses represent AIC-constrained multi-Gaussian fits showing three significant populations present in the sample. For an extended analysis, please refer to the Appendix.

of MTB in an environmental sample containing a mixture of MTB and non-magnetic bacteria.

TEM images revealed a wide range of MTB morphologies, including cocci, rods, and spirilla, as well as diverse magnetosome crystal shapes: cuboctahedral, bullet-shaped, prismatic, and "ladyfinger"-shaped. Different magnetosome configurations were found: single, double, and multiple chains, as well as disordered structures. The presence of multiple MTB morphotypes was further supported by 16S rRNA analysis, which identified members of three families known to contain MTB: *Sphingomonadaceae*, *Burkholderia-ceae*, and *Magnetococcaceae*.

To analyze MTB behavior at the single-cell level, we developed and applied two open-source characterization methods: a low-equipment velocimetry pipeline based on standard optical microscopy and a static MF, and a modified U-turn method for estimating magnetic moments in an alternating MF. These methods enabled the identification and separation of distinct MTB populations based on cell size, swimming velocity, and magnetic moment.

Velocimetry yielded velocity-radius histograms for two consecutive days and, combined with AIC-constrained multi-Gaussian fitting, revealed that four distinct populations were present in the sample on day 1. On day 2, only two main populations remained, with overall reduced mobility. These results suggest that MTB from the Ogre River are highly sensitive to changes in the environment, which may explain their rapid decline under *ex situ* conditions. Although velocimetry is a great tool for initial

**TABLE 3** Population parameters for the *m* data set (Fig. 11)

| $\mu_1$ ($m$, $10^{-15}\,A \cdot m^2$) | $\mu_2$ ($r_{\text{bacteria}}$, µm) | $\sigma_1$ | $\sigma_2$ | $\rho$ |
|---|---|---|---|---|
| $1.314 \pm 0.003$ | $0.784 \pm 0.001$ | $0.413 \pm 0.003$ | $0.062 \pm 0.001$ | $0.60 \pm 0.01$ |
| $1.63 \pm 0.01$ | $0.9017 \pm 0.0003$ | $0.49 \pm 0.01$ | $0.0279 \pm 0.0003$ | $-0.26 \pm 0.02$ |
| $2.35 \pm 0.01$ | $0.941 \pm 0.001$ | $0.99 \pm 0.01$ | $0.099 \pm 0.001$ | $0.816 \pm 0.004$ |

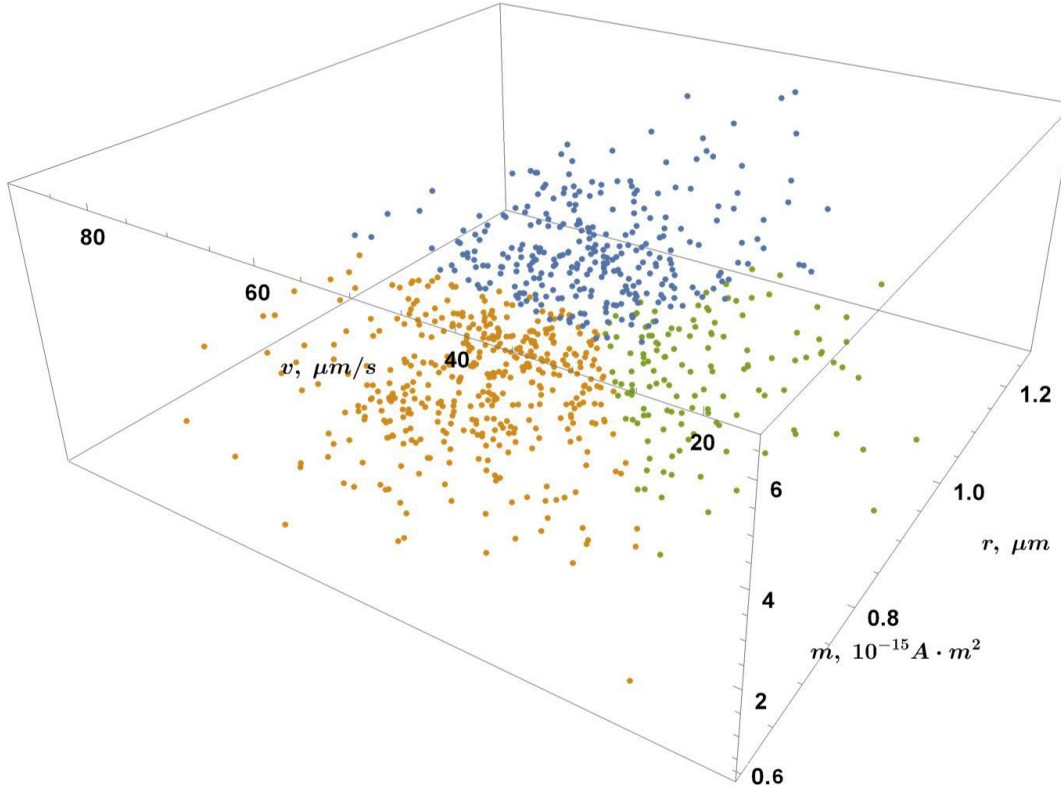

**FIG 12** Population analysis in the (*r, v, m*) MTB parameter space using variational Gaussian mixture clustering analysis. The three populations are colored in blue, yellow, and green.

population estimation in a natural sample, it takes into account all moving bacteria, including non-magnetic ones, which may result in the detection of additional populations.

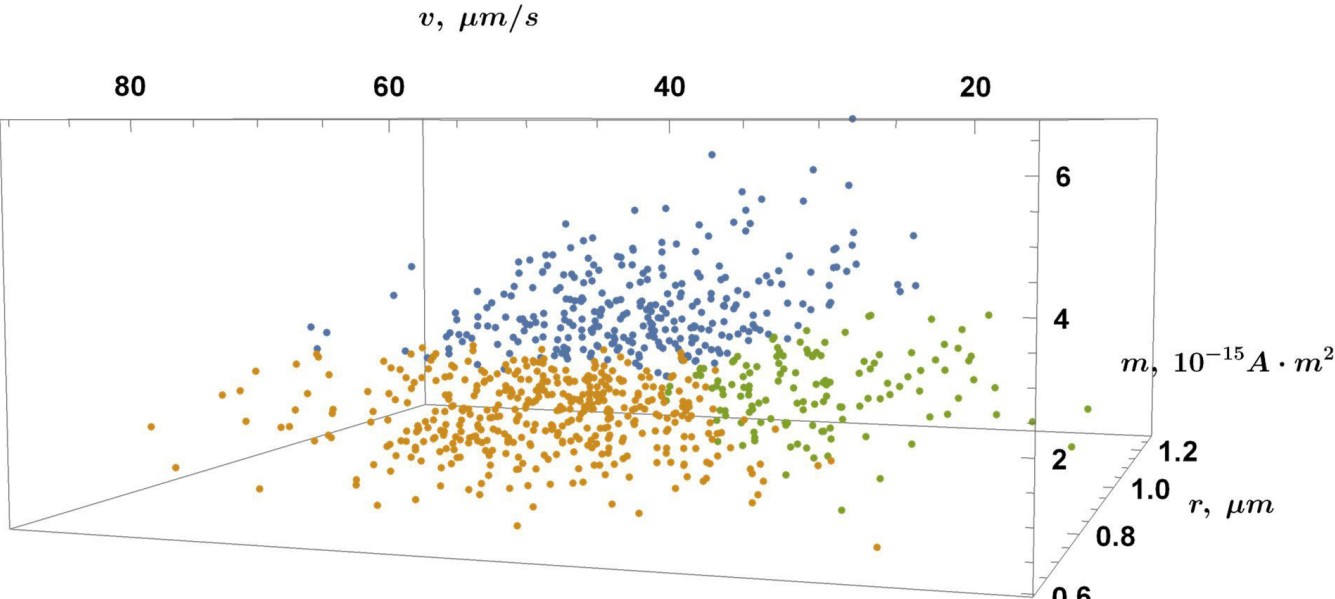

**FIG 13** Population analysis in the (*r, v, m*) parameter space: an additional point of view for clarity.

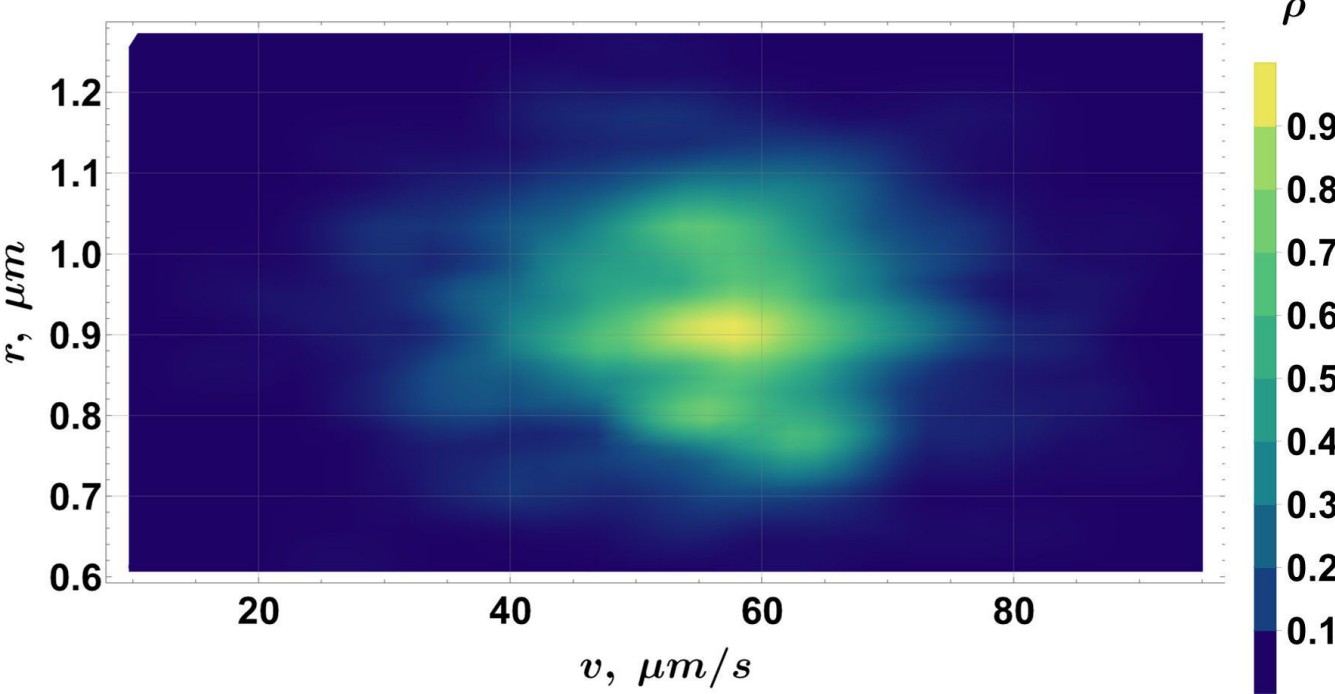

FIG 14 A smooth density histogram for the (r, v) projection of the data in the (r, v, m) space.

The U-turn method, while demanding a microscope equipped with an alternating MF generator, can be used to further distinguish between populations based on individual cell magnetic moments. We derived magnetic moment statistics from 846 trajectories, as well as put the data through the previously established velocimetry analysis pipeline. Although velocimetry identified two populations, analysis of the same data set,

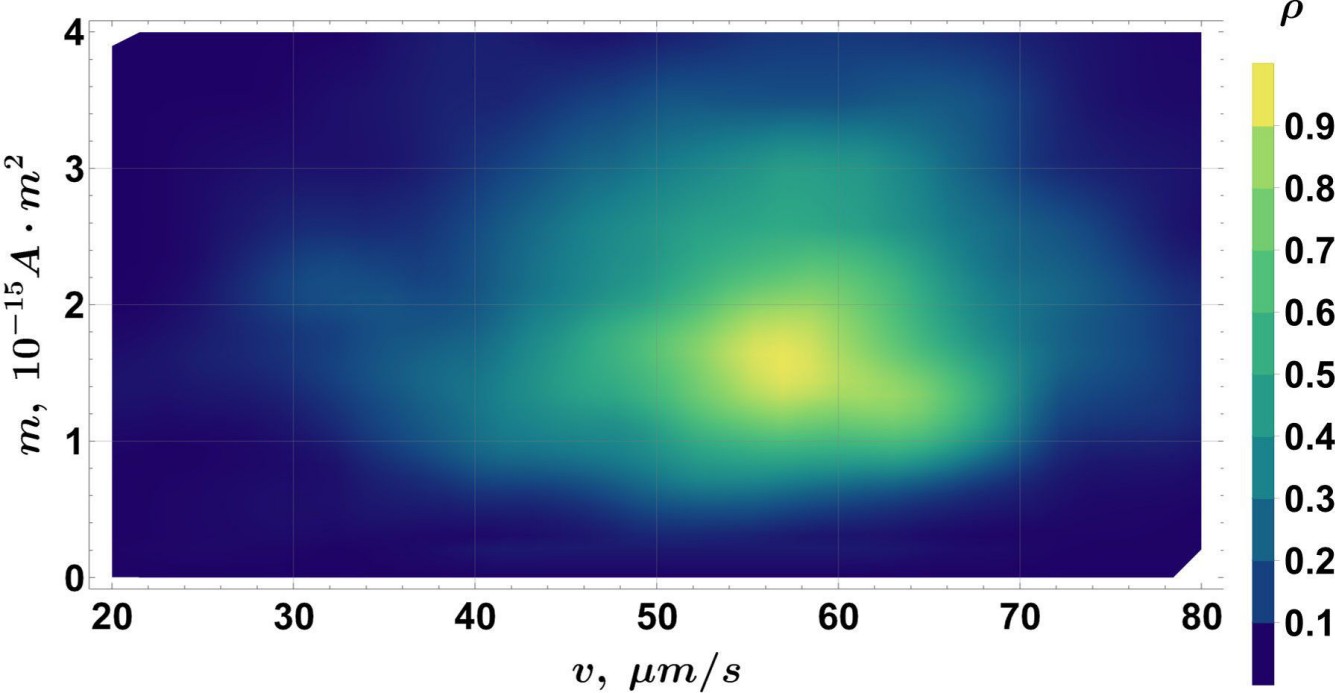

FIG 15 A smooth density histogram for the (m, v) projection of the data in the (r, v, m) space.

correlating magnetic moment and cell radius and using AIC constrained multi-Gaussian fitting, found three MTB populations in the sample. One of the velocimetry populations was found to be non-magnetic, bringing the total to four discernible groups.

Although AIC provides a statistically robust framework for model selection, it does not impose biophysical constraints and therefore does not prohibit populations with negative $m(r)$ correlations. In our analysis, we allowed for negative correlations, as the resulting three-population model provided the best fit according to AIC. Although such negative correlations would be biologically unusual, their presence in the data could reflect underlying species-level differences or uncharacterized magnetosome configurations. Another important consideration when performing population analysis of a natural sample is that different individual bacteria may have been recorded in each data set, due to the heterogeneous and uncharacterized composition of the natural microbial community.

The main goal of this study was to test the performance of recently developed image-based analysis methods in a previously uncharacterized environmental sample and then compare the obtained data with well-established methods used for MTB diversity research: TEM imaging and gene sequencing. In general, this study demonstrates that the integration of automated velocimetry and magnetic moment analysis provides a powerful and accessible approach to differentiating between MTB populations.

Furthermore, the data obtained provide a preliminary view of the considerable diversity of MTB found in the Ogre River. The coexistence of multiple MTB populations with different cell morphologies, swimming velocities, and magnetic moments suggests that these microorganisms may occupy different ecological niches within the same environment. This variation could reflect adaptations to local gradients in oxygen concentration, iron availability, or redox potential.

This work demonstrates the potential of image-based characterization as a tool for MTB diversity research. However, the applicability of these methods to other environmental conditions and MTB communities remains to be demonstrated in future studies.

## APPENDIX

### Population analysis in the ($r$, $v$, $m$) space

To further validate the three-population hypothesis suggested by the 2D multi-Gaussian fitting (Fig. 11), a clustering analysis was performed for the magnetic moment data set using a variational Gaussian mixture model (74) in the ($r$, $v$, $m$) MTB parameter space. Although such clustering can provide insight into the population structure of the sample, we favor the 2D Gaussian linear combination constrained by AIC over ($r$, $m$), as it accounts for possible population overlaps. In addition, in this case, there is not enough data for a 3D Gaussian linear combination fitting in the ($r$, $v$, $m$) space; hence, we use the strict clustering method for sparse data. As seen in Fig. 12 and 13, three groups (colored blue, yellow, and green) can be distinguished based on the magnetic moment, radius, and velocity of the cell, similar to Fig. 11.

While a correlation between cell magnetic moment and its size has been shown here, it can also be informative to examine the smooth density histograms of the two ($r$, $v$, $m$) data projections: ($r$, $v$) (Fig. 14) and ($m$, $v$) (Fig. 15). In Fig. 14, multiple groups can be distinguished according to their velocity and radius. Note that the global number density maximum in Fig. 14 corresponds to the population with $k \sim vk \in$ [18; 39] μm/s and $r \in$ [0.56; 0.81] μm in Fig. 6a (Table 2), despite the bias introduced by the filtering criteria applied in the moment calculation algorithm. No correlation can be found between $m$ and $v$ in Fig. 15, as expected, since there is currently no evidence suggesting that the magnetic moment depends on the velocity of the cell.

## ACKNOWLEDGMENTS

The authors express their gratitude to Bhagyashri Shinde, Martins Klevs, Janis Cimurs and his children, and Malo Marmol for participating in the expedition to acquire bacteria samples. D.F. thanks M. Floriani for access to the IRSN TEM.

M.S., M.B., A.C., and G.K. acknowledge funding from the Latvian Council of Science, project A4Mswim, project Nr. lzp-2021/1-0470. M.S., A.K., J.L., and G.K. acknowledge the LLC "MikroTik" donation project (no.2319), administered by the University of Latvia foundation, "Magnetotactic bacteria in nature and applications (MTBna)." J.L. acknowledges funding from the Latvian Council of Science, project lzp-2021/1-0522. The authors are grateful to the French-Latvian bilateral program "Osmose," project Nr. LV-FR/2023/3. D.F. acknowledges the BioMagnetLink project (grant agreement ID: 101187789).

## AUTHOR AFFILIATIONS

[1]MMML lab, Department of Physics, University of Latvia, Riga, Latvia
[2]CEA, CNRS, BIAM, Aix-Marseille Université Saint-Paul-lez-Durance, Marseille, France
[3]Institute of Microbiology and Biotechnology, University of Latvia, Riga, Latvia
[4]Latvian Biomedical Research and Study Centre, Latvia, Riga
[5]HUN-REN-PE Environmental Mineralogy Research Group, Research Institute of Biomolecular and Chemical Engineering, University of Pannonia, Veszprém, Hungary
[6]Department of Physics, University of Latvia, Riga, Latvia

## AUTHOR ORCIDs

Mara Smite http://orcid.org/0000-0003-3410-787X
Agnese Kokina http://orcid.org/0000-0001-6043-1986

## AUTHOR CONTRIBUTIONS

Mara Smite, Conceptualization, Data curation, Formal analysis, Investigation, Methodology, Validation, Visualization, Writing – original draft, Writing – review and editing | Mihails Birjukovs, Conceptualization, Data curation, Formal analysis, Methodology, Software, Validation, Visualization, Writing – original draft, Writing – review and editing | Mila Sirinelli-Kojadinovich, Data curation, Formal analysis, Methodology | Sandrine Grosse, Data curation, Methodology | Agnese Kokina, Data curation | Janis Liepins, Formal analysis, Writing – review and editing | Dita Gudra, Data curation | Megija Lunge, Data curation | Davids Fridmanis, Formal analysis, Writing – review and editing | Mihaly Posfai, Visualization, Writing – review and editing | Andrejs Cebers, Data curation, Writing – review and editing | Damien Faivre, Conceptualization, Supervision, Visualization, Writing – review and editing | Guntars Kitenbergs, Conceptualization, Data curation, Formal analysis, Funding acquisition, Project administration, Resources, Supervision, Writing – review and editing

## DATA AVAILABILITY

Raw sequence data have been deposited at the European Nucleotide Archive under study accession no. PRJEB101386. The code and data (archived) associated with this article can be downloaded at Zenodo: https://zenodo.org/records/17339197. Repository contents are as follows: images obtained in velocimetry and magnetic moment measurement experiments with magnetotactic bacteria (MTB); code used to detect MTB from images, reconstruct MTB trajectories, and determine their magnetic moment, size, and velocity statistics (Wolfram Mathematica and Python); code output: MTB properties and trajectory data, as well as relevant statistics in the form of .pdf and .svg images, .csv and compressed .txt files, .pkl and .wxf binary formats, in addition to videos, other figures, and code snapshots in .pdf format. In addition to the Zenodo repository, the image analysis, object tracking, and trajectory analysis code for $m$ calculations are available on GitHub: MTB detection from images, Mihails-Birjukovs/

MTB_detection_tracking; MTB tracking, Peteris-Zvejnieks/MHT-X; MTB magnetic moment retrieval from trajectories, Mihails-Birjukovs/MTB_magnetic_moment_from_U-turn_trajectories.

## ADDITIONAL FILES

The following material is available online.

### Supplemental Material

**Video S1 (Spectrum02563-25-s0001.mp4).** Ogre River MTB in alternating magnetic field, trajectories used to calculate magnetic moment.
**Video S2 (Spectrum02563-25-s0002.mp4).** MTB velocimetry in static magnetic field, day 1.
**Video S3 (Spectrum02563-25-s0003.avi).** MTB velocimetry in static magnetic field, day 2.

### Open Peer Review

**PEER REVIEW HISTORY (review-history.pdf).** An accounting of the reviewer comments and feedback.

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
