## [Reviewer comments · Microbiology Spectrum]

Microbiology Spectrum

Image-based physical characterization of magnetotactic bacteria from an environmental sample

Mara Smite, Mihails Birjukovs, Mila Sirinelli-Kojadinovich, Sandrine Grosse, Agnese Kokina, Janis Liepins, Dita Gudra, Megija Lunge, Davids Fridmanis, Mihály Pósfai, Damien Faivre, Andrejs Cebers, and Guntars Kitenbergs

Corresponding Author(s): Mara Smite, University of Latvia

Review Timeline:

Submission Date:	September 3, 2025
Editorial Decision:	October 8, 2025
Revision Received:	November 21, 2025
Accepted:	January 26, 2026

Editor: Blaire Steven

Reviewer(s): The reviewers have opted to remain anonymous.

Transaction Report:

DOI: <https://doi.org/10.1128/spectrum.02563-25>

Re: Spectrum02563-25 (Image-based physical characterization of magnetotactic bacteria from an environmental sample)

Dear Dr. Mara Smite:

Thank you for the privilege of reviewing your work. Below you will find my comments, instructions from the Spectrum editorial office, and the reviewer comments.

Overall, I found the manuscript quite interesting and am happy to report the reviewers felt the same. After a few minor modifications I believe we will be able to accept the manuscript. Please pay particular attention to the need for data to be deposited in a recognized repository.

Revision Guidelines

Sincerely,
Blair Steven
Editor
Microbiology Spectrum

Reviewer #1 (Comments for the Author):

作者利用基于图像的物理表征，结合16S rRNA测序和TEM形态学分析。对食人魔河环境样本中的趋磁细菌（MTB）进行了系统研究。作者开发了开源的自动化图像分析管道，包括基于光学显微镜的细胞测速和磁矩测量，可以以较低的设备成本实现MTB群落

的定量分析。这项工作和方法论上具有很强的创新性，与期刊的主题相符。对笔者的一些评论：1. 图 1 应按照科学制图惯例重新绘制。建议包括坐标网格、指北针（指南针）和比例尺，以提高空间精度和地理参考的清晰度。2. 本研究的核心创新在于开发了自动化图像处理和磁矩测量方法。然而，缺乏与标准手动测量技术的比较验证。鼓励作者将手动获得的结果与从自动化算法得出的结果进行直接比较，并讨论潜在的误差来源，例如照明异质性、跟踪不准确或磁场不稳定性。3. 采样结果是否具有代表性？结果是否受到采样季节或沉积环境的影响？4. 鞘氨醇单胞菌属的相对丰度达到 94.9%，这对于富集磁性的微生物样品来说异常高。作者应该详细讨论鞘氨醇单胞菌在这种情况下可能的起源，并评估其优势是否会干扰或偏向磁响应数据的解释。5. 作者应明确描述磁场是如何产生、测量和验证的，以确保实验的可重复性和物理可靠性。6. 目前，讨论部分更多地侧重于技术描述，缺乏对生物学意义的深入解释。7. 建议作者检查全文：（1）加强数据的一致性和相互关系，（2）通过额外的相关文献加强论证，以及（3）确保所有结论都得到所提出证据的充分支持。

Reviewer #2 (Comments for the Author):

#Study Summary:

Magnetotactic bacteria (MTB) exhibit the remarkable ability to navigate by combining passive magnetic alignment to Earth's magnetic field via magnetosomes with flagella-driven swimming and chemo-/aerotactic responses. However, our understanding of the remarkable diversity of MTB found in environmental samples is still limited due to the difficulty of isolating them in axenic culture, which has so far been achieved only for a few species. The present study introduces a novel, integrated methodology that combines 16S rRNA sequencing, transmission electron microscopy, and automated image-based physical approaches (tracking) to characterize mixed bacterial populations (including MTB, but also non-MTB) from environmental samples. The study specifically targets a newly identified MTB-rich site - the Ogre River - which was the only location among three sampled freshwater environments in Latvia found to contain MTB. Sampling from the Ogre River revealed a highly diverse MTB population, including various cell morphologies and magnetosome structures. Moreover, genetic analysis based on 16S rRNA sequencing showed high taxonomic diversity, while motility behavior analysis demonstrated rapid post-sampling changes in motility within 48 hours, indicating environmental sensitivity of the MTB populations with a rapid post-sampling decline in viability.

#Major Comments:

This work presents an interesting approach to MTB population characterization. The automated, image-based physical methods are adoptable and could be helpful for future environmental sample assessments. Coupled with cell morphological and genetic analysis, the paper offers a multidimensional view of microbial ecology and behavior. That said, application to other environmental conditions and MTB communities remains to be demonstrated. This is at least something the authors might conclude at the end of their "Conclusions & Outlook". I am aware that this is beyond the scope of a single study; therefore, I suggest addressing it as a potential direction for future work. Overall, the manuscript is well written and I have only a few, mostly text edit-based suggestions that could be considered to further improve clarity and completeness. Please find my suggestions below:

#Specific Comments:

Line 22 (and any other places where "16S rRNA sequencing" appears in the text): Wouldn't the more appropriate term be 16S rDNA sequencing, since the method targets the gene encoding the 16S rRNA rather than the RNA molecule itself?

Lines 44-45: I suggest rephrasing to reflect that magnetosome chains are generally observed, but that random arrangements have also been reported in the literature. This would provide a more balanced description of known magnetosome organization patterns. As the topic of random organization is addressed later (Lines 62-63), it may be also acceptable to refer to chains here as the typical arrangement, while acknowledging variability in Line 62-63 in the manuscript.

Line 134: Please indicate the intensity of the magnetic field produced by the "weak permanent" magnet used in the experiments.

Line 134-137: Please explain why South-seeking behavior was expected. Since the samples originate from the Northern Hemisphere, one would expect North-seeking behavior - i.e., predominant accumulation of cells at the edge of drop facing a magnet's south pole under atmospheric conditions. This matches the behavior described in the manuscript, suggesting that the terminology may have been used incorrectly. The terms "North-seeking" and "South-seeking" can be confusing because they do not indicate migration relative to the physical magnetic field direction, but rather the predominant polarity bias with respect to the geographic origin of the bacteria. Given the current orientation of Earth's geomagnetic field, the North Geomagnetic Pole actually corresponds to a physical magnetic south pole, and vice versa. North-seeking bacteria in the Northern Hemisphere and South-seeking MTB in the Southern Hemisphere both swim preferentially downward within an oxygen gradient when exposed to excess oxygen - toward physical magnetic south (parallel to B) in the former case and toward physical magnetic north (antiparallel to B) in the latter.

Line 278: "bilopotrichous" should read "bilophotrichous."

Line 326: Please define what is meant by "effective radii." Since the authors are targeting a microbiology journal, it would be helpful to explain the biophysical parameters in more detail.

Line 344: Please clarify whether a velocity of approximately 2 $\mu\text{m/s}$ can still be considered active swimming, or if it should be

interpreted as passive movement of MTB.

Lines 472-474: Consider using "natural" as a synonym for "wild."

Fig. 2a: I'm not sure whether the small cell attached to the spirillum qualifies as a separate "smaller MTB." It could also be a mini-cell from the same strain, possibly resulting from (impaired) cell division.

Fig. 2b: The caption should read "bullet-shaped magnetosomes" instead of "bullet-shape magnetosomes."

Fig. 2c: "Cell aggregate" would be a more appropriate description than "cluster."

Fig. 2e and Line 285: Not sure whether "disordered" is fully accurate - perhaps "partly disordered" would be more appropriate, as a chain-like arrangement is still visible. Moreover, in the chain-like arrangement, it is interesting to note that the magnetosome crystals are oriented with their long axes perpendicular to the direction of the chain.

Table 2: Please include information on sample sizes. "radus" should probably read "radius."

Supplement: Consider including a supplemental video showing the swimming behavior of the observed MTB and the tracking method used.

The author utilized image-based physical characterization, combined with 16S rRNA sequencing and TEM morphological analysis. A systematic study was conducted on magnetotactic bacteria (MTB) in the environmental samples of the Ogre River. The author has developed an open-source automated image analysis pipeline, including cell velocimetry and magnetic moment measurement based on optical microscopes, which can achieve quantitative analysis of MTB communities at low equipment costs. This work is highly innovative in methodology and aligns with the journal's theme. However, there are still certain deficiencies in the current manuscript. Overall, the article has the potential for publication, but it requires significant revisions and data supplementation.

Some comments for the author:

1. Figure 1 should be redrawn following scientific mapping conventions. It is recommended to include coordinate grids, a north arrow (compass), and a scale bar to improve spatial accuracy and clarity of geographic reference.
2. The core innovation of this study lies in the development of automated image processing and magnetic moment measurement methods. However, comparative validation with standard manual measurement techniques is lacking. The authors are encouraged to provide a direct comparison between manually obtained results and those derived from the automated algorithms, and to discuss potential sources of error, such as illumination heterogeneity, tracking inaccuracies, or magnetic field instability.
3. Are the sampling results representative? Is the result affected by the sampling season or the sedimentary environment?
4. The relative abundance of the genus *Sphingomonas* reaches 94.9%, which is unusually high for a magnetically enriched microbial sample. The authors should discuss in detail the possible origin of *Sphingomonas* in this context and evaluate whether its dominance may interfere with or bias the interpretation of the magnetic response data.
5. The authors should include a clear description of how the magnetic field was generated, measured, and validated to ensure reproducibility and physical reliability of the experiments.
6. Currently, the discussion section focuses more on technical descriptions and lacks in-depth explanations of biological significance.

7. It is suggested that the author check the full text: (1) strengthen data consistency and interrelationships, (2) enhance argumentation through additional relevant literature, and (3) ensure all conclusions are well-supported by the presented evidence.

Image-based physical characterization of magnetotactic bacteria from an environmental sample

by Mara Smite et al.

Dear Referee(-s) and Editor(-s),

Below are the comments/questions received from Reviewer 2, as well as our responses. Please note that, for convenience, there is also a separate .pdf file with highlighted revisions with respect to the original draft.

#Study Summary:

Magnetotactic bacteria (MTB) exhibit the remarkable ability to navigate by combining passive magnetic alignment to Earth's magnetic field via magnetosomes with flagella-driven swimming and chemo-/aerotactic responses. However, our understanding of the remarkable diversity of MTB found in environmental samples is still limited due to the difficulty of isolating them in axenic culture, which has so far been achieved only for a few species. The present study introduces a novel, integrated methodology that combines 16S rRNA sequencing, transmission electron microscopy, and automated image-based physical approaches (tracking) to characterize mixed bacterial populations (including MTB, but also non-MTB) from environmental samples. The study specifically targets a newly identified MTB-rich site - the Ogre River - which was the only location among three sampled freshwater environments in Latvia found to contain MTB. Sampling from the Ogre River revealed a highly diverse MTB population, including various cell morphologies and magnetosome structures. Moreover, genetic analysis based on 16S rRNA sequencing showed high taxonomic diversity, while motility behavior analysis demonstrated rapid post-sampling changes in motility within 48 hours, indicating environmental sensitivity of the MTB populations with a rapid post-sampling decline in viability.

#Major Comments:

This work presents an interesting approach to MTB population characterization. The automated, image-based physical methods are adoptable and could be helpful for future environmental sample assessments. Coupled with cell morphological and genetic analysis, the paper offers a multidimensional view of microbial ecology and behavior. That said, application to other environmental conditions and MTB communities remains to be demonstrated. This is at least something the authors might conclude at the end of their "Conclusions & Outlook". I am aware that this is beyond the scope of a single study; therefore, I suggest addressing it as a potential direction for future work. Overall, the manuscript is well written and I have only a few, mostly text edit-based suggestions that could be considered to further improve clarity and completeness. Please find my suggestions below:

We thank the Referee for taking time to thoroughly read the manuscript and for the valuable feedback. Every suggestion has been taken into account and used to further improve the manuscript.

The main goal of this paper was to demonstrate how novel image processing methods can be used to characterize a natural sample, as a community of unknown number of MTB species is significantly more difficult to describe than a simple single culture grown in a lab. We then compared the imaged based methods to well established ways that are used in MTB research: gene sequencing and TEM imaging. As a result, a significant portion of the manuscript focuses on technical details, but we have tried to add additional analysis that explores the biological significance of the results in more depth.

We completely agree that the results of this study do not demonstrate a complete picture of microbial ecology of Ogre River. It is now clarified in the manuscript that the emphasis is placed on novel image-based techniques that can be used in tandem with more established methods, or independently to characterize a complex biological sample. To have a better understanding of Ogre River MTB population composition and motility behaviour, major additional work should be carried out in the future, including sampling at different seasons and collecting statistics on how the genetic diversity changes over time.

#Specific Comments:

Line 22 (and any other places where "16S rRNA sequencing" appears in the text): Wouldn't the more appropriate term be 16S rDNA sequencing, since the method targets the gene encoding the 16S rRNA rather than the RNA molecule itself?

In principle, both terms can be used. *16S rRNA V3-4 gene sequencing* would be the most exact term in this case, as 16S rRNA gene variable region V3 and V4 is being sequenced. This gene encodes ribosomal RNS, which is the ribosomal subunit in prokaryotes. Some authors prefer to use *16S rDNA sequencing*, however in the majority of publications in the fields of molecular biology, microbiome and others the term *16S rRNA gene sequencing* is used, as that is the gene name that is being targeted by primers.

→ We would prefer that the term 16S rRNA sequencing remains in the manuscript unchanged, as it is used more widely than 16S rDNA sequencing. However, in places where 16S rRNA sequencing is discussed, it is now clarified that we are talking about 16S

rRNA gene sequencing.

Lines 44-45: I suggest rephrasing to reflect that magnetosome chains are generally observed, but that random arrangements have also been reported in the literature. This would provide a more balanced description of known magnetosome organization patterns. As the topic of random organization is addressed later (Lines 62-63), it may be also acceptable to refer to chains here as the typical arrangement, while acknowledging variability in Line 62-63 in the manuscript.

→ It has been changed to "The magnetosomes are typically ordered in chains along the easy axis of magnetization, which corresponds to the direction of magnetic moment."

Line 134: Please indicate the intensity of the magnetic field produced by the "weak permanent" magnet used in the experiments.

A 1.2 T neodymium magnet was used in the field expeditions to concentrate the MTB in sample jars and to extract them later, as well as in experiments with light microscopy to confirm presence of MTB in the sites that were sampled. The magnet was placed a few centimeters from the jars/ microscope stage and therefore the resulting magnetic field that interacted with the cells was decreasing rapidly (estimated to be less than 10 miltesla). We did a quick calculation to estimate the magnetic field strength B at distance r from the sides of the cube shaped magnet :

r, cm	B, mT
2.5	7.0549
2.75	5.55285
3.	4.4481
3.25	3.61771
3.5	2.98164
3.75	2.48626
4.	2.09476
4.25	1.78129
4.5	1.52736
4.75	1.31947
5.	1.14765
5.25	1.00441
5.5	0.884046
5.75	0.782167
6.	0.695358

In this context the field was considered to be "weak" as in strong enough to attract MTB, but not so strong as to damage the cells.

→ The magnet strength and the expected field strength at the sample has been specified in the manuscript.

Line 134-137: Please explain why South-seeking behavior was expected. Since the samples originate from the Northern Hemisphere, one would expect North-seeking behavior - i.e., predominant accumulation of cells at the edge of drop facing a magnet's south pole under atmospheric conditions. This matches the behavior described in the manuscript, suggesting that the terminology may have been used incorrectly. The terms "North-seeking" and "South-seeking" can be confusing because they do not indicate migration relative to the physical magnetic field direction, but rather the predominant polarity bias with respect to the geographic origin of the bacteria. Given the current orientation of Earth's geomagnetic field, the North Geomagnetic Pole actually corresponds to a physical magnetic south pole, and vice versa. North-seeking bacteria in the Northern Hemisphere and South-seeking MTB in the Southern Hemisphere both swim preferentially downward within an oxygen gradient when exposed to excess oxygen - toward physical magnetic south (parallel to B) in the former case and toward physical magnetic north (antiparallel to B) in the latter.

Thank you for catching this mistake, indeed it should read North-seeking (seeking the magnetic south).

→ It has been corrected in the manuscript and an additional figure has been added to demonstrate this behaviour and make the manuscript easier to read.

Line 278: "bilopotrichous" should read "bilophotrichous."

→ It has been corrected in the manuscript.

Line 326: Please define what is meant by "effective radii." Since the authors are targeting a microbiology journal, it would be helpful to explain the biophysical parameters in more detail.

→ The concept and calculation of effective radius has been explained more in depth in the manuscript, under the "Magnetic moment calculation" subsection.

Line 344: Please clarify whether a velocity of approximately 2 $\mu\text{m/s}$ can still be considered active swimming, or if it should be interpreted as passive movement of MTB.

Indeed, velocities from 7 $\mu\text{m/s}$ or more are considered as active swimming for bacteria. It is known, that Magnetococci can achieve swimming velocities of 100 $\mu\text{m/s}$ or more (Bente et al., 2020), therefore, yes, we agree, that "2 $\mu\text{m/s}$ " could be coined as "passive movement of MTB".

Bente, K., Mohammadinejad, S., Charsooghi, M.A., Bachmann, F., Codutti, A., Lefèvre, C.T., Klumpp, S. and Faivre, D., 2020. High-speed motility originates from cooperatively pushing and pulling flagella bundles in bilophotrichous bacteria. *Elife*, 9, p.e47551.

→ It has been noted in the manuscript, that anything under 7 $\mu\text{m/s}$ is passive movement.

Lines 472-474: Consider using "natural" as a synonym for "wild."

→ "Wild" has been replaced with "natural" in the manuscript.

Fig. 2a: I'm not sure whether the small cell attached to the spirillum qualifies as a separate "smaller MTB." It could also be a mini-cell from the same strain, possibly resulting from (impaired) cell division.

It is impossible to say what exactly these little "cells" with magnetosomes are without further research. We assumed that they are separate cells, as they were observed in multiple TEM images - not just the one that was included in the manuscript. Here we have attached two more TEM images, where the attached "cells" seem to have different magnetosome morphology than the larger MTB cells they are attached to.

At this point we can only speculate if it would be possible for cells to mutate so much, that proteins that regulate cell division and the proteins that regulate magnetosome formation have both been changed at the same time.

→ The possibility that it could be a minicell is included in the manuscript, as well as a new reference to publication that researches cell division in MSR-1.

Fig. 2b: The caption should read "bullet-shaped magnetosomes" instead of "bullet-shape magnetosomes."

→ It has been corrected in the manuscript.

Fig. 2c: "Cell aggregate" would be a more appropriate description than "cluster."

→ Word "cluster" has been replaced with "cell aggregate" in the caption and where appropriate in the text.

Fig. 2e and Line 285: Not sure whether "disordered" is fully accurate - perhaps "partly disordered" would be more appropriate, as a chain-like arrangement is still visible. Moreover, in the chain-like arrangement, it is interesting to note that the magnetosome crystals are oriented with their long axes perpendicular to the direction of the chain.

→ "partly disordered" is now used instead of "disordered".

Table 2: Please include information on sample sizes. "radus" should probably read "radius."

→ The typo in the Table 2 caption has been corrected. The sample size for each of the subfigures in Figure 6 (and the metrics in Table 2) is ~75K instances of radius/velocity pairs. Sample size has been included in the manuscript.

Supplement: Consider including a supplemental video showing the swimming behavior of the observed MTB and the tracking method used.

→ The videos have been added to the submission.

Image-based physical characterization of magnetotactic bacteria from an environmental sample

by Mara Smite et al.

Dear Referee(-s) and Editor(-s),

Below are the comments/questions received from Reviewer 1, as well as our responses. Please note that, for convenience, there is also a separate .pdf file with highlighted revisions with respect to the original draft.

The author utilized image-based physical characterization, combined with 16S rRNA sequencing and TEM morphological analysis. A systematic study was conducted on magnetotactic bacteria (MTB) in the environmental samples of the Ogre River. The author has developed an open-source automated image analysis pipeline, including cell velocimetry and magnetic moment measurement based on optical microscopes, which can achieve quantitative analysis of MTB communities at low equipment costs. This work is highly innovative in methodology and aligns with the journal's theme. However, there are still certain deficiencies in the current manuscript. Overall, the article has the potential for publication, but it requires significant revisions and data supplementation.

We thank the Referee for the feedback and will do our best to incorporate the suggestions.

Some comments for the author :

1. Figure 1 should be redrawn following scientific mapping conventions. It is recommended to include coordinate grids, a north arrow (compass), and a scale bar to improve spatial accuracy and clarity of geographic reference.

The figure has been updated as requested.

2. The core innovation of this study lies in the development of automated image processing and magnetic moment measurement methods. However, comparative validation with standard manual measurement techniques is lacking. The authors are encouraged to provide a direct comparison between manually obtained results and those derived from the automated algorithms, and to discuss potential sources of error, such as illumination heterogeneity, tracking inaccuracies, or magnetic field instability.

Regarding the validation against standard manual measurement techniques - the reason why we developed a fully-automated solution to MTB detection, tracking and property measurements is because errors cannot be reliably quantified for manual measurements. Some estimates may be given, but they cannot be rigorously verified. Consider the example of a 2018 paper by Pichel et al. "Magnetic response of *Magnetospirillum gryphiswaldense* observed inside a microfluidic channel" (<https://doi.org/10.1016/j.jmmm.2018.04.004>) – here, a semi-manual magnetic moment measurement method relies on manual and rather arbitrary segmentation of bacteria U-turns entirely by hand. As we've established, U-turn segmentation affects the characteristic velocity computed for the U-turn, as well as the "observed" U-turn asymptotic width. However, since there is no criterion for manually selecting the U-turn fragments, errors cannot be even estimated. At the very least, nothing along these lines is presented. There are similar issues with the other published semi-automatic methods.

Next, about errors due to illumination heterogeneity, tracking inaccuracies, or magnetic field instability. Inhomogeneous illumination has virtually no effect on the detection quality, since it is compensated for by the flat-field correction during image processing, in addition to image histogram-adaptive contrast-to-noise boosting and maximum filters that are, after all previous corrections, luminance gradient-agnostic. We outline the procedure in our previous paper (DOI: 10.1016/j.bpj.2025.06.008). The image processing and tracking codes also automatically detect and correct for field of view shifts during experiments, can handle out-of-focus bacteria, as well as variable focus/reflections and diffraction artifacts (Airy patterns). The bacteria detection and tracking methods have been thoroughly validated in the context of turbulent and multiphase flow analysis. This is described in detail in a series of papers:

<https://doi.org/10.1007/s00348-022-03399-5>

<https://doi.org/10.1007/s00348-022-03445-2>

<https://doi.org/10.1007/s00348-024-03793-1>

<https://doi.org/10.1103/PhysRevFluids.5.061601>

https://doi.org/10.3390/app11209710?urlappend=%3Futm_source%3Dresearchgate

<https://doi.org/10.1007/s00348-023-03671-2>

<https://doi.org/10.1371/journal.pone.0322069>

<https://doi.org/10.1103/73gp-gdsw>

The tracking and the improved U-turn methods have been described in more detail in a recent publication by the authors,

where the magnetic moment (m) of MSR-1 is studied in depth (DOI: [10.1016/j.bj.2025.06.008](https://doi.org/10.1016/j.bj.2025.06.008)). Here, two methods for measuring the magnetic moment are compared: the time-based method, which is commonly used for manual measurements, and the newly proposed method which is based on the distance between the two branches of the trajectory.

The new method is designed to be robust, and includes multiple filtering steps, to ensure that only true U-turns survive to the stage where they are used to calculate m :

- 1) Removing trajectories that do not sufficiently conform to the magnetic field dynamics;
- 2) Removing self-intersecting trajectory segments;
- 3) Removing trajectories where cells move out of focus plane, i.e. the U-turns are not 2-dimensional;
- 4) Multiple filtering stages that ensure that trajectories are correctly decomposed into symmetrical U-turns;
- 5) The magnetic moment is computed not based on the time steps elapsed for a U-turn, which is prone to artifacts at lower frame rates (most of the experiments reported in literature), whereas in our case a theoretical U-turn shape function is fitted to the experimentally determined U-turn, and the moment is computed from the characteristic U-turn velocity (automatically measured for the reversion region of the U-turn, detected automatically as well) and the asymptotic width L of the U-turn.

Validation of our method against the conventional method clearly showed our approach is more robust and less sensitive to the imaging conditions, plus it does not rely on arbitrary U-turn segmentation, since that is built into the analytical model for the U-turn shape that we've developed – our model is essentially a modified Bean model, which recasts the U-turn-like motion into a more explicit form suitable for MTB motion and property characterization.

In this article, our aim was to expand the new method beyond a single lab grown culture, and see how it will perform with a more complex sample.

3. Are the sampling results representative? Is the result affected by the sampling season or the sedimentary environment?

The sampling was performed at the exact same location two times, and is representative of the current microbiome state, not the MTB ecosystem in the Ogre River as whole.

The sampling was performed in summer and autumn, which means that the water level and temperature was different in both cases. In summer, the temperature was about 20 degrees more than autumn, and the water level was about 1 meter higher in the fall. This directly impacts the sampling depth, as it was significantly harder to obtain sediment samples in the fall.

This study focuses more on the methods that can be used to characterize a complex sample, than a through characterization of the MTB populations and their dynamics through seasonal changes. For that, multiple expeditions during all four seasons should be performed and the data correlation studied.

→ A note has been included in the methods section of the manuscript briefly explaining this.

4. The relative abundance of the genus *Sphingomonas* reaches 94.9%, which is unusually high for a magnetically enriched microbial sample. The authors should discuss in detail the possible origin of *Sphingomonas* in this context and evaluate whether its dominance may interfere with or bias the interpretation of the magnetic response data.

Genus *Sphingomonas* consists of species highly abundant in the soil and water. They have been found in the rivers before (Tabata et al. 1999). Also *Sphingomonas* spp. have a tendency to form biofilms together with other microorganisms (Bolhuis, et al, 2011). While we did not characterise the species abundance of magnetically non-enriched river bed samples, we presume that initial sampling has disrupted a microbial mat and over time (48 h) motile bacteria could migrate out from it and evenly diffuse throughout the sample.

When quantifying sample species abundance, we relied on the molecular methods (DNA extraction, PCR and Illumina sequencing) which are not biased to species richness per se. From the point of magnetic response, most probably *Sphingomonas* spp. form the population with velocity of $0 - 10^1$ μm and radius approx 0.5, see figure 8 (Jiang et al., 2023)

Henk Bolhuis, Lucas J Stal, Analysis of bacterial and archaeal diversity in coastal microbial mats using massive parallel 16S rRNA gene tag sequencing, The ISME Journal, Volume 5, Issue 11, November 2011, Pages 1701–1712, <https://doi.org/10.1038/ismej.2011.52>

Tabata K, Kasuya K, Abe H, Masuda K, Doi Y. 1999. Poly(Aspartic Acid) Degradation by a *Sphingomonas* sp. Isolated from Freshwater. *Appl Environ Microbiol* 65: Poly(Aspartic Acid) Degradation by a *Sphingomonas* sp. Isolated from Freshwater | Applied and Environmental Microbiology

Jiang L, Choe H, Peng Y, Jeon D, Cho D, Jiang Y, Lee JH, Kim CY, Lee J. 2023. *Sphingomonas abietis* sp. nov., an Endophytic Bacterium Isolated from Korean Fir. *J Microbiol Biotechnol.* ;33(10):1292-1298. doi: 10.4014/jmb.2303.03017

→ We have included this new information in the manuscript, and added the references.

5. The authors should include a clear description of how the magnetic field was generated, measured, and validated to ensure reproducibility and physical reliability of the experiments.

The field is measured indirectly, and is controlled via data acquisition card and custom Labview program. It has been validated that 1 ampere provided to the coil pair will create a 1.7 mT strong magnetic field at the capillary with bacteria suspension. We do not calibrate the device before every experiment, as the device has proved itself to be robust and precise - in a publication from MMML group where the same device is used, the magnetic field was measured and controlled using a triple axis compass magnetometer sensor module, which showed that the magnetic field can be defined with 0.03 mT precision (M. Brics et. al., 2023, DOI: <https://doi.org/10.1103/PhysRevE.108.024601>).

→ An image of experimental setup has been added, as well as a more thorough explanation of how the magnetic field is generated and controlled.

6. Currently, the discussion section focuses more on technical descriptions and lacks in-depth explanations of biological significance.

The primary objective of the study was to test the performance of recently developed image processing methods within a complex, previously uncharacterized sample: specialized cell tracking algorithm (MHT-X), improved and automated magnetic moment measurement and cell velocimetry in static magnetic field. As a result, the focus was on validating the robustness and applicability of the techniques, which involves a great deal of technical details. However, we do agree that the biological implications of our findings are as important.

→To address this, we have

- Speculated on the possible origins of the cell aggregates and minicells in the morphological and genetic diversity section
- Analyzed the genetic diversity in more detail
- Expanded the discussion section to briefly interpret the observed diversity of MTB morphologies, magnetic moments and motility behaviours in the context of ecology. These additions provide biological context, while maintaining the technical focus of the manuscript.

A more detailed investigation of the MTB community in the Ogre River would require multiple sampling campaigns in different seasons and additional genomic and image based analyses, which are planned for future studies.

7. It is suggested that the author check the full text: (1) strengthen data consistency and interrelationships, (2) enhance argumentation through additional relevant literature, and (3) ensure all conclusions are well-supported by the presented evidence

We thank the reviewer for this constructive recommendation. We have revised the manuscript, and all changes in the revision are highlighted for convenience. Specifically:

1. We reviewed all figures and tables, and some captions have been revised to be more precise. Additional details have been clarified in the text, methods used explained more thoroughly and the work reframed slightly to focus on the novel methods used for complex environmental sample characterization, as was the goal of the study from the beginning.
2. Additional references have been added to the bibliography, providing additional information on the experimental setup, MTB detection methods, MTB ecology and biology.
3. The conclusions section has been revised to more clearly state the scope of the study.

Re: Spectrum02563-25R1 (Image-based physical characterization of magnetotactic bacteria from an environmental sample)

Dear Dr. Mara Smite:

Your manuscript has been accepted, and I am forwarding it to the ASM production staff for publication. Your paper will first be checked to make sure all elements meet the technical requirements. ASM staff will contact you if anything needs to be revised before copyediting and production can begin. Otherwise, you will be notified when your proofs are ready to be viewed.

Sincerely,
Blair Steven
Editor
Microbiology Spectrum